# Understanding the interplay between urban segregation and accessibility to services with network analysis

Mirko Lai[1☯], Anna Sapienza[1☯], Salvatore Vilella[1], Massimo Canonico[1], Federica Cena[2‡], Giancarlo Ruffo[1‡*]

**1** Dipartimento di Scienze e Innovazione Tecnologica, Università del Piemonte Orientale, Alessandria, Italy, **2** Dipartimento di Informatica, Università degli Studi di Torino, Turin, Italy

☯ These authors contributed equally to this work.
‡ FC and GR are joint senior authors, and also contributed equally to this work.
* giancarlo.ruffo@uniupo.it

## Abstract

The 15-minute city concept has gained momentum as an urban planning strategy to enhance livability, inclusiveness, and sustainability by ensuring that essential services are within a short walk or bike ride from home. However, while hyper-proximity is often promoted as desirable, its potential side effects on spatial segregation and social exclusion remain underexplored. In this paper, we propose a network-based analytical framework to investigate whether hyper-proximity models — such as the 15-minute city — may inadvertently reinforce spatial segregation by shaping service accessibility and urban transport connectivity. We model cities as complex spatial networks, quantify accessibility using the distribution of relevant Points of Interest (PoIs) and employ closeness centrality as a proxy for connectivity across multiple scales, from residential addresses to network-derived clusters and entire cities. Our results show that areas with better access to services generally exhibit higher connectivity and vice versa. However, this tendency is uneven when looking at socio-demographic factors. Some neighborhoods, particularly lower-income ones, experience both lower accessibility and weaker connectivity. In contrast, certain higher-income neighborhoods display low accessibility and limited connectivity, suggesting patterns of voluntary isolation. These findings indicate that hyper-proximity alone does not guarantee inclusiveness and may mask underlying socio-economic inequalities.

## Introduction

The 15-minute city concept represents a significant urban planning strategy aimed at enhancing city livability and inclusiveness [1,2]. Grounded in the hyper-proximity paradigm, it envisions neighborhoods in which residents can access essential services and amenities within a 15-minute walk or bicycle ride from their homes.

**Data availability statement:** The data underlying the results presented in the study are available from GitHub (https://github.com/mirkolai/cities).

**Funding:** This research has been partially funded by PON "Ricerca e Innovazione" 2014-2020 716 framework, and by the European Union - Next Generation EU, Mission 4 Component 2 717 - CUP C53D23005810006, and CUP C33C24000340001. There was no additional external funding received for this study. The funders had no role in study design, data collection and analysis, decision to publish, or preparation of the manuscript.

**Competing interests:** NO authors have competing interests.

This approach seeks to reduce car dependency, promote healthier lifestyles, and improve environmental sustainability.

In recent years, substantial advances have been made in computational methods and tools for assessing and promoting hyper-proximity in urban environments [3,4]. A wide range of metrics has been proposed to quantify city walkability and service accessibility [5–9], enabling the development of analytical platforms, such as the CityChrone Project [10], the City Access Map [11], and The X Minute City [12].

However, this urban design paradigm also raises important challenges. While improving local access to services is a key objective, such interventions may inadvertently contribute to gentrification processes and increased socio-spatial segregation [13]. As a result, recent research has emphasized that accessibility assessments should move beyond purely spatial or temporal proximity measures to account for additional dimensions, including diversity in activity timing, perceived accessibility, and fairness in distribution of opportunities across different population groups and urban areas [14–16]. In this context, Lucas et al. [17] highlight the need to integrate ethical considerations with technical metrics, stressing their implications for social inclusion, participation, and community cohesion. Similarly, Van Wee [18] conceptualizes accessibility-related equity issues as arising either from insufficient overall access or from unjust disparities between social groups.

Within this broader debate, transport and land-use systems play a central role, as they directly shape both the level and the spatial distribution of accessibility. Understanding how these systems interact is therefore crucial for evaluating the inclusiveness of accessibility-oriented planning strategies. Our work contributes to this ongoing discussion by adopting multi-dimensional indicators that capture not only proximity and functional diversity of services, but also public transport connectivity. This integrated approach enables a data-driven, connectedness-aware assessment of service accessibility and supports the identification of urban areas where limited access to efficient urban transport may increase the risk of spatial exclusion, as observed in previous studies [19–21]. Our main objectives, outlined in the next section, thus focus on understanding the relationship between urban transport connectivity and accessibility to services.

## Objectives and scope

To investigate the relationship between hyper-proximity fitness and urban connectivity patterns, we present an analytical framework based on complex network theory. We focus on two key dimensions: accessibility and closeness, analyzed over multiple scales, i.e., cities across the globe, citywide, neighborhood, and individual residential address levels. We represent the urban landscape as a directed graph, where nodes are street intersections and edges are road segments connecting them [22,23]. Then, we measure accessibility to Points of Interest (PoIs) through walkable or bikable edges, as in [11,24]. To calculate closeness, we instead analyze the urban transport network (as in [10,25]), where the edges in the graph must be served by buses, metros, local trains, etc. Additionally, we categorize cities into neighborhoods by clustering intersections using the Infomap algorithm [26] and study their closeness centrality

to identify segregated neighborhoods. Finally, by leveraging publicly available socio-demographic data for Italian cities, we provide insights into the heterogeneous distribution of these metrics, highlighting the complex relationship between service accessibility and connectivity, while identifying urban areas where one or more of these measures are notably low.

By quantifying accessibility and closeness across neighborhoods, our framework not only characterizes urban connectivity patterns but also provides the basis for evaluating equity and, as such, is explicitly linked to the spatial justice agenda [27,28], underscoring the importance of equitable access to urban resources as a foundational principle of just cities. Spatial justice is not a replacement for social or economic justice but a complementary perspective that highlights the spatial dimensions inherent in all forms of justice and injustice. By operationalizing concepts such as distributive and procedural justice, the framework not only reveals existing inequalities but also provides a road-map for corrective action. Integrating these metrics into urban policy supports the creation of more inclusive, resilient, and sustainable urban environments, aligning with broader goals of fairness and opportunity for all residents. Moreover, this approach provides policymakers with practical tools to identify transit inequities and guide interventions that promote spatial justice. Such interventions may include revising zoning policies, enhancing service coverage, and preventing transit clustering.

Finally, it is worth noting that urban studies have long explored the relationships between service accessibility and other important dimensions, including mobility patterns, land value and use, housing characteristics, and taxation. These factors affect urban form, density, sprawl, and development dynamics, influencing housing supply, investment, and economic growth in complex ways (e.g., [29]), ultimately influencing the quality of the services at different urban scales. While acknowledging the importance of these interactions, a comprehensive analysis encompassing all such dimensions lies beyond the scope of this study.

## Data

Our study relies on multiple data sources. We first select cities based on the combined lists analyzed by Bruno et. al [25] and Nicoletti et. al [11], and add the 10 most populous Italian cities. This results in a total of 92 cities worldwide.

We then collect the following information for each city. We use *OpenStreetMap* (OSM), a free crowd-sourced map of the world, to retrieve the pedestrian road network and the PoI positions. To access and process these geospatial data, we rely mainly on the Python libraries `osmnx` and `geopandas`. In addition to computing various spatial metrics from the OSM data, we also produce visual map representations. These are generated from the geographic coordinates of the intersections and the road segments connecting them, using `matplotlib` for plotting.

While OSM data is known to have heterogeneous coverage across regions, it remains a valuable open data source enabling comparative analyses across a wide range of cities. These data allow us to compute paths and measure spatial and temporal distances between nodes (i.e., their residents) and nearest services or amenities, as well as between different neighborhoods within a city.

To assess accessibility to services from the perspective of a single point, we first need to assign a service category to each Point of Interest (PoI). Although there is no consensus among scholars on which categories to use or how to group services, we base our classification of a PoI's category on the one proposed by Nicoletti et al. [11]. In particular, we map the list of OSM tags to a higher-level category of services. When a tag is no longer available or has been replaced, we select the most similar related alternative. This classification includes seven categories: *Mobility*, *Active Living*, *Entertainment*, *Food Choices*, *Community Space*, *Education* and *Health and Well-being*. We refer to this set of categories as CAT.

Given a city $c$, and the set of overall PoIs in $c$ $\text{POI}_c = \{p_1, p_2, \ldots\}$, we map each PoI $p \in \text{POI}_c$ to one of the categories with notation $\text{cat}(p)$. More information on the mapping of the PoIs tags to main categories can be found in the Supplementary Material S1 Appendix: Point of Interests' OSM categories.

Since detailed residential data (e.g., number of residents, residential addresses, etc.) is often incomplete or unavailable, we adopt a simplified approach. Instead of using actual residential addresses, we use road network intersections as proxies for population locations, as they provide homogeneous coverage of urban areas. To estimate the resident

population at each intersection, we used WorldPop (https://www.worldpop.org/, last access: Jan. 22, 2026), a global population dataset providing estimated population densities at fine spatial resolution (i.e., 100-meter grid cells). We then evenly distribute the estimated resident population within each grid cell across the intersections located within that cell.

Additionally, we retrieve public transport schedules in GTFS format, when available, from the Transitland service (https://www.transit.land/, last access: Jan. 22, 2026), selecting feeds that fall within each city's bounding box. For Italian cities not covered by Transitland, as well as for Istanbul, we collected GTFS data from open data sources provided by local administrations or official public transport websites. Public transport data were collected for a total of 81 cities out of 92. In the paper, we show the analysis on these 81 cities to fairly compare accessibility to service and overall connectivity. However, we include the accessibility analysis for the remaining 11 cities in the Supplementary Material (S2 Appendix: More on cities' rankings and S3 Appendix: Distributions of $\mathcal{P}(n)$, $\mathcal{D}(15, n)$, $\mathcal{E}(15, n)$, and $\mathcal{A}(15, n)$ for all the other cities).

Finally, for Italian cities only, we collect official data from the Italian Ministry of Economy and Finance on average resident income for the year 2022 (https://tinyurl.com/bdcww24t, last access: Jan. 22, 2026). These data are publicly available, grouped by postal code (CAP). However, since official postal code boundaries in Italy are proprietary and not publicly accessible, we estimate them by constructing concave hulls based on the distribution of addresses extracted from OpenStreetMap. The average income for each intersection is then assigned according to the postal code area it falls within.

## Methods

Fig 1 and the following list summarize the multi-stage approach used to achieve the research objectives.

**Data Collection:** Multiple datasets were assembled for 92 cities globally, including pedestrian road networks and Points of Interest (PoIs) from OpenStreetMap, population estimates from WorldPop, and public transport schedules in GTFS format from Transitland or local sources. For Italian cities, average resident income data by postal code were also collected to allow socio-economic correlations.

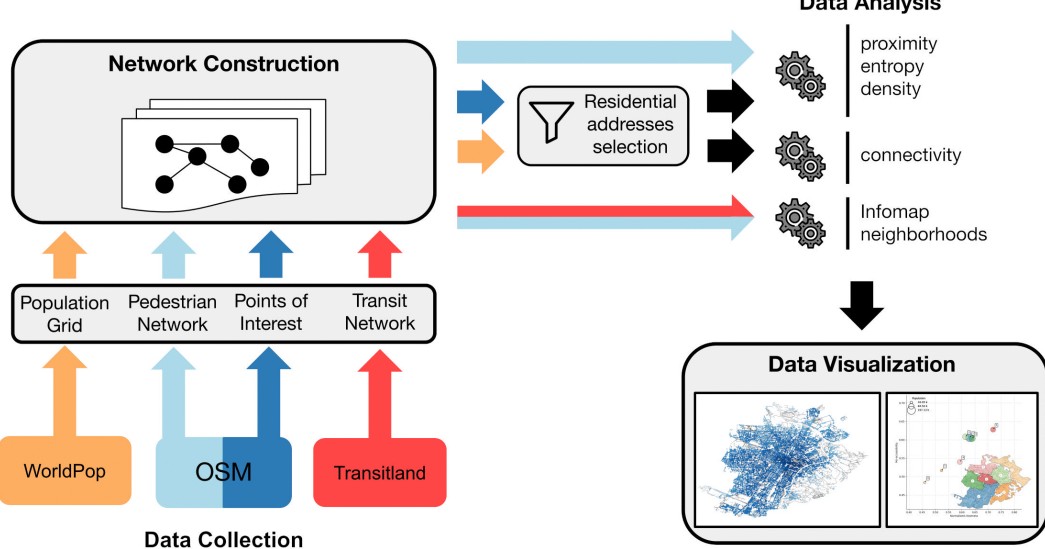

**Fig 1. Workflow diagram illustrating the sequential stages of the research methodology.** Data from various sources were integrated to analyze service accessibility and connectivity, visualized to explore spatial patterns across urban areas.

**Network Construction:** The pedestrian network was modeled as a weighted, directed graph where nodes represent street intersections and edges represent walkable street segments, weighted by walking distance and time. PoIs and residential population were mapped onto the closest nodes, using intersection proxies for residences. An urban transport network graph was created by integrating public transit routes and schedules into the pedestrian graph.

**Data Analysis:** Node-level metrics of PoI-proximity, PoI-density, and PoI-entropy were computed for each intersection to quantify service accessibility within walkable distances for all residential addresses (i.e., street intersections within populated WorldPop cells). Closeness centrality was calculated on the urban transport network to measure each node's connectivity to the entire city. These measures were aggregated at multiple scales, including census blocks, administrate districts, and neighborhoods detected by the Infomap community detection algorithm.

**Data Visualization:** Graphical representations were created to visually explore and study the spatial distribution of service accessibility, connectivity, and socio-demographic patterns across the urban areas.

### PoI's proximity, density, entropy as node-wise measures

For each city $c$, we build a *pedestrian network*. This is a weighted directed graph $G_c^{\text{ped}} = (N_c, E_c)$, where each node $n \in N_c$ is an intersection and an edge $e_{ij} \in E_c : e_{ij} = (n_i, n_j)$ is a walkable street segment between nodes $n_i$ and $n_j$. Every PoI is assigned to the closest node in $G_c^{\text{ped}}$ with the function $\text{poi}(p) = n_p$, that maps each $p \in \text{POI}_c$ to a node $n_p \in N_c$. Similarly, we assign every residential address to the closest node in the pedestrian network. We weight edges by the distance in meters from one endpoint to the other $d(e_{ij})$, and the average time $t(e_{ij})$ to walk from $n_i$ to $n_j$.

Given two nodes $n, m \in N_c$, they are connected if there exists a path in $G_c^{\text{ped}}$. Hence, we define the length of the minimum shortest path in meters between $n$ and $m$ as the *spatial distance* $d(n, m)$, and the length of the minimum shortest path in minutes between $n$ and $m$ as the *time distance* $t(n, m)$. In general, we use $d(x, y)$ and $t(x, y)$ to indicate the spatial and temporal distance between any two elements $x$ and $y$, which can be intersections, PoIs, or residential addresses.

This representation allows us to calculate the walkable distance, both in meters and in time travel (assuming an average velocity of 5 km/h), between any two points in a city. We note that it is possible to improve our model by storing the actual distances, in meters and walking time, from each PoI $p \in \text{POI}_c$ and each residential address to the node $n$ they were assigned to. However, in this paper, we simplify the analysis by not considering these short-range distances.

We re-interpreted the most common definitions of the 15-minute city concept (see [1]) to quantitative network metrics to get a three-dimensional estimate of accessibility to services (see [30] for a systematic categorization of various approaches to proximity assessment).

First, let's recall that the proximity dimension should assess if basic services are readily available to residents within a 15-min radius.

**Definition 1 (PoI-proximity)** *Given a node $n \in N_c$, the PoI-proximity $\mathcal{P}(n)$ is the minimum temporal radius needed by residents in $n$ to reach at least one PoI for each category in $\mathrm{CAT}$. Thus, $\mathcal{P}(n)$ is the minimum temporal distance such that $\forall x \in CAT, \exists p \in POI_c : (cat(p) = x \land \mathcal{P}(n) \le t(n, p))$.*

For example, if $\mathcal{P}(n) = 10$, as in Fig 2, then residents assigned to $n$ will find at least one PoI in each category within 10 minutes. Note that this temporal distance is minimal, i.e., at least one category is not accessible through walks shorter than 10 minutes.

An alternative method for estimating proximity to PoIs from different categories is to average the distances from the nodes to their nearest service in each category. For example, the average walkable distance from $n$ to PoIs $i$, $a$, and $x$ in Fig 2 is 7.33 minutes, instead of *10*. Although we prefer to use the upper bound value, as defined in Def. 1, to estimate worst case scenarios, other studies, such as those by Nicoletti et al. [11], and Bruno et al. [25], have used the average. In the rest of the paper, we refer to this alternative formulation of the PoI-proximity with the notation $\mathcal{P}_{\text{avg}}(n)$.

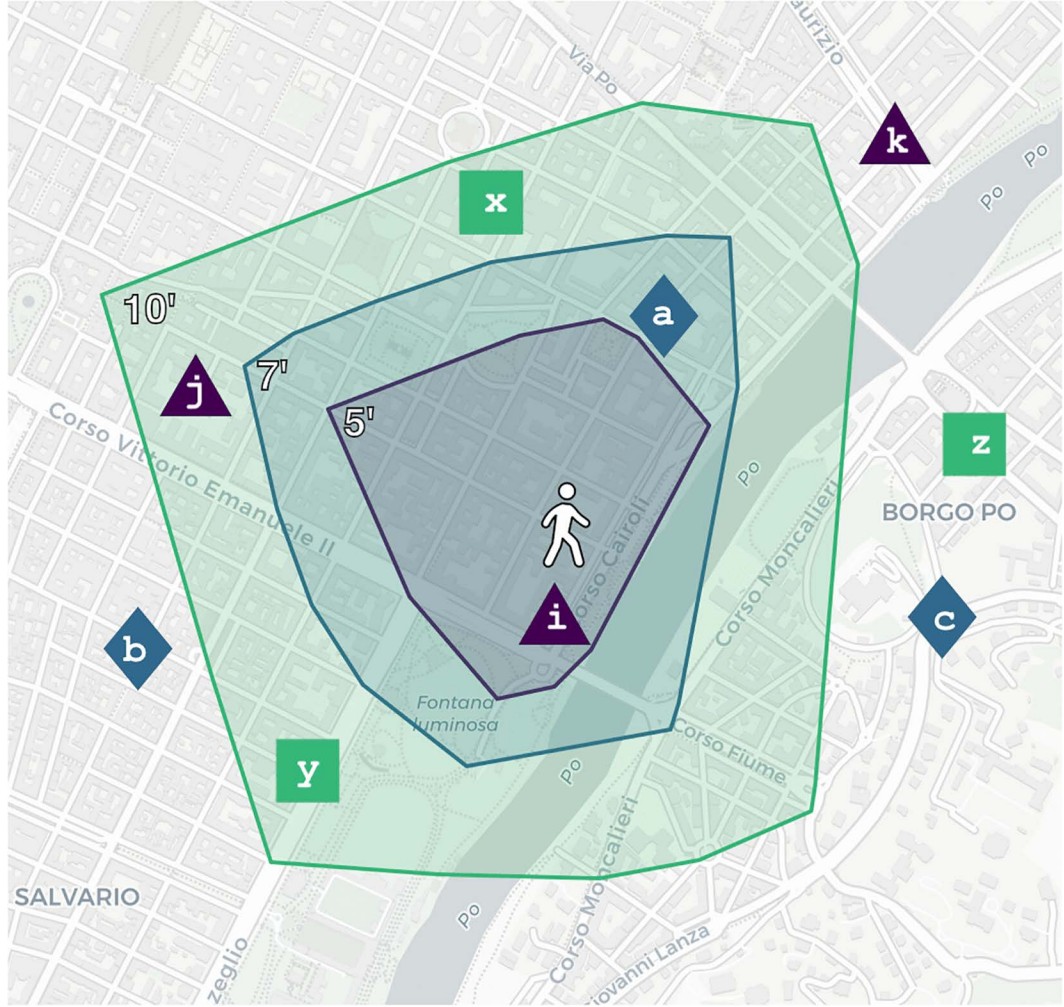

**Fig 2. Example of a node *n* and its closest Points of Interest (PoIs).** Three example service categories are shown: purple triangles, blue diamonds, and green squares. The closest PoIs in each category are respectively *i* (at walkable distance of 5 minutes), *a* (at 7 minutes), and *x* (at 10 minutes). In this example, the PoI-proximity $\mathcal{P}(n) = 10$ because residents in *n* can walk to a PoI for each of the given categories within 10 minutes. Equivalently, the largest isochrone for *n* that includes at least one service in each category is $\mathcal{I}(10, n)$. *Base map and data from OpenStreetMap and OpenStreetMap Foundation.*

**Definition 2 (PoI-proximity-avg)** *Given a node $n \in N_c$, the PoI-proximity-avg $\mathcal{P}_{\mathrm{avg}}(n)$ is the average distance needed by residents in n to reach at least one PoI for every category in CAT.*

A comparison of our proximity-based city ranking with the other studies mentioned above is briefly discussed in the Experimental analysis section.

PoI-proximity returns the de-facto 'hyper-proximity' time parameter for each node in the graph. For example, if $\mathcal{P}(n) \leq 15$ for every node *n* in $N_c$, one might be tempted to call *c* a 'perfect' 15-minute city. However, this is not entirely accurate, as $\mathcal{P}(n)$ alone is not enough to draw meaningful conclusions. For instance, a neighborhood might only have one PoI for each category within walking distance, which can be problematic for two main reasons: 1) the number of services in the area should be proportional to the number of residents living nearby; and 2) every category in CAT can include a quite heterogeneous number of tags, corresponding to very different kind of services, e.g., both 'hospital' and 'pharmacy' are members of the 'Health and Well-being'

class according to OSM. Therefore, we propose the use of PoI-proximity alongside other measures. Once an area around a given node $n$ has been identified, we estimate how many PoIs are present inside that area (i.e., PoI-density), and also how many PoIs of different categories are reachable within a walk of $t$ minutes from $n$ (i.e., PoI-entropy).

To define these additional measures, we make use of an isochrone map, which describes a region accessible from a given geographical point. These maps have been used since 1881 [31], and are made of contours marking the area reachable within a certain time threshold (see an illustrative example in Fig 2).

**Definition 3 (Isochrone)** *Given a node $n \in N_c$, the isochrone $\mathcal{I}(t, n)$ is a geographical area whose internal points are walkable from $n$ in at most $t$ minutes. If $t = \mathcal{P}(n)$, we simply refer to the isochrone of $n$ with $\mathcal{I}(n)$.*

In conventional urban planning, density is often associated with the concentration of buildings in urbanized areas. In contrast, within the 15-minute city framework, Allam et al. [2] conceptualize density as the optimal number of services needed to sustain the resident population within a given urban area. Here, we propose a new metric (PoI-density) which quantifies density as the ratio between the number of Points of Interest (PoIs) reachable within a 15-minute walking isochrone and the area of that isochrone, expressed in square kilometers.

**Definition 4 (PoI-density)** *Given a node $n \in N_c$ and a time $t$ in minutes, the PoI-density $\mathcal{D}(t, n)$ is the ratio between the number of PoIs in isochrone $\mathcal{I}(t, n)$ and the area in square kilometer of $\mathcal{I}(t, n)$. If $t = \mathcal{P}(n)$, we simply refer to the PoI-density of $n$ with $\mathcal{P}(n)$.*

It is possible that, for certain regions, the number of PoIs exceeds the area in square kilometers. To simplify comparisons and to keep $\mathcal{D}(t, n)$ in the range of [0, 1], we apply a min-max normalization.

Finally, we measure how PoIs in a given isochrone are distributed across categories.

**Definition 5 (PoI-entropy)** *Given a node $n \in N_c$ and a time $t$ in minutes, the PoI-entropy $\mathcal{E}(t, n) = -\sum_{x \in CAT}(Pr(x) \cdot \log Pr(x))$, where $Pr(x)$ is the probability that a PoI $p$ within the isochrone $\mathcal{I}(t, n)$ belongs to category $x$ (i.e., $cat(p) = x$).*

High entropy generally indicates a mixed-use area with a variety of amenities and services, while low entropy tends to indicate more specialized zones, such as residential or industrial areas.

Note that our definition of entropy overlaps with the measure of diversity used in Moreno et al. [1]. In their work, diversity is defined as a twofold concept: 1) the presence of mixed-use neighborhoods; and 2) socio-cultural diversity among residents. However, due to the difficulty of obtaining socio-demographic data at fine spatial resolutions, we focus solely on entropy to quantify the diversity of accessible services.

While socio-demographic data can offer valuable insights into aspects of accessibility to services, our analysis focuses on three distinct measures — PoI-proximity, PoI-density, and PoI-entropy — that together provide a more comprehensive view of accessibility. In contrast to previous studies, such as those by [10] and [11], which typically rely on a single measure, our approach leverages multiple metrics to capture different dimensions of accessibility, offering a more nuanced analytical tool for urban planning.

Our three measures are complementary from a semantic point of view. We can aggregate these measures into a single linear combination that returns a combined **PoI-accessibility** score of a node $n$ at time $t$:

$$\mathcal{A}(t, n) = w_{\mathcal{P}}\mathcal{P}'(n) + w_{\mathcal{D}}\mathcal{D}'(t, n) + w_{\mathcal{E}}\mathcal{E}'(t, n) \tag{1}$$

where $w_{\mathcal{P}}$, $w_{\mathcal{D}}$, and $w_{\mathcal{E}}$ are the weights that can be attributed to the corresponding measure, $t$ is set to a fixed time in minutes, while the apex indicates that the measures have been normalized by *min-max* scaling. If we set $t = \mathcal{P}(n)$, we can simply define the PoI-accessibility of $n$ as $\mathcal{A}(n) = w_{\mathcal{P}}\mathcal{P}'(n) + w_{\mathcal{D}}\mathcal{D}'(n) + w_{\mathcal{E}}\mathcal{E}'(n)$. In this study, we set $w_{\mathcal{P}} = w_{\mathcal{D}} = w_{\mathcal{E}} = 1/3$, and $t = 15$.

To define a general measure where values approach 0 when accessibility is low and 1 when accessibility is high, we normalize PoI-proximity differently from PoI-density and PoI-entropy. Higher PoI-proximity indeed indicates that one or more PoIs of certain categories are further from a given intersection, which should worsen the overall accessibility score. As a result, the normalized PoI-proximity $\mathcal{P}'$ is defined as it follows:

$$\mathcal{P}'(n) = \frac{\max_n \mathcal{P}(n) - \mathcal{P}(n)}{\max_n \mathcal{P}(n) - \min_n \mathcal{P}(n)} \tag{2}$$

The combined accessibility score provides an overall evaluation of service reachability in a city. However, as our analysis shows, it is still important to consider the individual metrics of proximity, density and entropy to explore weaknesses in urban planning in more detail.

**Using closeness centrality to measure distances and connectivity within a city**

While our accessibility measures help assess whether amenities and services are within walking distance, we are also interested in understanding how well each point is connected to the rest of the city, allowing residents to easily reach other urban areas and people.

To capture this second aspect, we build the *urban transport network*, a weighted directed graph $G^{\text{urb}}_c = (N_c, L_c)$, where $c$ is a city and each link $l_{ij} \in L_c : l_{ij} = (n_i, n_j)$ is a street segment between two nodes $n_i$ and $n_j$ served by urban public transport (e.g., buses, metro, local trains). We use GTFS feeds from Transitland to map transit stops to the nearest OSM nodes in $G^{\text{ped}}_c$ and we compute travel times between stops based on GTFS schedules.

The final $G^{\text{urb}}_c$ graph integrates both walking and transit edges, weighted by travel time, allowing us to study the connectivity of a node with respect to the rest of the city and, consequently, how isolation patterns may vary across the city. To this aim, we compute the closeness centrality score for each node in $G^{\text{urb}}_c$, which help identify how easily a resident can access the entire urban transport network from any given location.

**Definition 6 (closeness)** *The closeness of node n is defined as*

$$\mathcal{C}(n) = \sum_{m \neq n : m \in N_c} \frac{|N_c| - 1}{t(n, m)},$$

*where $t(n, m)$ is the temporal distance between n and m in the urban transport network $G^{\text{urb}}_c$.*

Note that we compute the temporal distance $t(n, m)$ as the average travel time across all scheduled trips on the same line and across the different lines serving the two nodes.

The above definition has been introduced by Beauchamp [32] as the reciprocal of the harmonic mean of distances, and it allows comparisons between graphs of different sizes and not necessarily connected. In our applicative domain, closeness centrality measures how close an intersection, and its related PoIs and residential addresses, is to all the other intersections in $c$. Higher values of closeness $\mathcal{C}(n)$ indicate that $n$ is well connected to the rest of the city, while lower values of $\mathcal{C}(n)$ indicate that $n$ is more topologically isolated with respect to other parts of the city.

To better capture disparities in connectivity across urban areas, we compare each node to the best connected ones in the city by normalizing closeness scores as follows:

$$\mathcal{C}'(n) = \frac{\mathcal{C}(n)}{\max_n \mathcal{C}(n)} \tag{3}$$

Thus, given $n \in N_c$, $\mathcal{C}'(n)$ measures $n$'s connectivity as a fraction of the closeness value of the best connected node in the city $c$. This normalization helps us better assess whether urban connectivity is evenly distributed across a city or, conversely, quantifies how spatially isolated the least connected areas are.

Note that we use closeness centrality as a proxy for urban segregation, in the specific sense of topological isolation from opportunities. However, we recognize that urban exclusion is a multidimensional, multilayered, and dynamic phenomenon, deeply embedded in social, economic, and political structures [33]. Closeness centrality, while useful for identifying network-based inaccessibility, cannot fully capture broader processes of marginalization, such as institutional bias,

affordability constraints, or lack of agency in mobility choices. As Lees [34] argues, spatially based mechanisms aiming at promoting social mixing do not necessarily translate into meaningful integration, and may coexist with persistent socioeconomic segregation. This reinforces the need to interpret structural measures like closeness within a broader understanding of urban inequality. Nonetheless, our metric (together with the other accessibility metrics) can be meaningfully situated within the framework of geographical exclusion and exclusion from facilities, as originally conceptualized by Church et al. [35], where spatial configuration and transport accessibility shape the unequal distribution of opportunities across the urban environment. Within this framing, closeness centrality provides a network-based perspective on spatial segregation, while acknowledging its limitations in capturing the full spectrum of urban exclusion.

Table 1 summarizes the previously defined node-wise measures for a quick reference.

## Measuring accessibility and connectivity for multi scale regions

We can extend the node-level metrics defined above to larger urban regions at different scales. For any given area, such as a census block, administrative district, network cluster, or the entire city, we are able to compute aggregated values by averaging PoI-proximity, PoI-density, PoI-entropy, and closeness centrality scores. In the following analysis, we focus on the distributions of these metrics, explore their descriptive statistics, and make use of bubble charts [36].

Bubble charts allow us to represent different urban regions, such as census areas, or entire neighborhoods, thus helping the identification of outliers, anomalies, and areas with lower accessibility and connectivity. Specifically, nodes located on the left side of the diagrams are more disconnected from other parts of the city, while those in the bottom section indicate lower accessibility to services. We also make use of the bubble size to visualize information such as the population of a given area, and the color code for categorical information, e.g., continent, administrative neighborhoods, etc. This analysis allows us to detect potential inequalities in service accessibility and connectivity both within and between cities.

Furthermore, the aggregated measures can be compared with socio-demographic data, such as income levels, to uncover possible patterns of urban exclusion. As a case study, we demonstrate this approach by analyzing the 10 most populated Italian cities, where we compare our metrics with the average income of residents grouped by postal code.

Official administrative districts boundaries are not always available or up to date for every city, making it challenging to rely on them for spatial analysis. To overcome this limitation, we identify 'natural' neighborhoods by running a community detection algorithm on the pedestrian network $G_c^{\text{ped}}$. Specifically, we apply the Infomap community detection algorithm [37], which is well-suited for detecting clusters in weighted and directed networks.

Infomap relies on the probability flow of random walks to partition the network into modules by minimizing the description length of the walker's trajectory. Conceptually, this models a pedestrian who, at each intersection, randomly chooses the next street to follow. As the walker moves through the network, tightly connected groups of nodes are traversed more

**Table 1. Summary of the accessibility and connectivity metrics used in the study.**

| Metric | Symbol | Description |
|---|---|---|
| Isochrone | $\mathcal{I}(t, n)$ | A geographical area walkable from node $n$ in at most $t$ minutes. |
| PoI-proximity | $\mathcal{P}(n)$ | Minimum time needed to reach at least one PoI for each category from $n$. |
| PoI-proximity-avg | $\mathcal{P}_{\text{avg}}(n)$ | Average distance to reach at least one PoI for every category from node $n$. |
| PoI-density | $\mathcal{D}(t, n)$ | Ratio of PoIs in isochrone $\mathcal{I}(t, n)$ to its area (PoIs per km$^2$). |
| PoI-entropy | $\mathcal{E}(t, n)$ | Distribution diversity of PoI categories within isochrone $\mathcal{I}(t, n)$. |
| Closeness | $\mathcal{C}(n)$ | How well connected a node $n$ is to all other nodes in the urban transport network. |

frequently, naturally revealing a city's neighborhood structure. As observed in [38], this method offers a city-agnostic way to detect neighborhoods without depending on potentially outdated administrative boundaries.

## Computational challenges and the Cloud Computing infrastructure

The computational cost of our methodology is influenced by several factors: the number of nodes, the scarcity of PoIs, and the memory usage for storing PoI coordinates. On the one hand, we compute a node's closeness score by using an exact algorithm [39], where the primary factor influencing the computational workload is the number of nodes. On the other hand, previous works computed proximity by dividing cities into fixed-size grid cells, whereas our approach reduces approximation by considering individual intersections, significantly increasing the number of computation points. Moreover, rather than exclusively computing proximity based on 15-minute isochrones, we dynamically determine isochrones ranging from 1 to 60 minutes, stopping only when a PoI from every category is found. This flexibility in isochrone computation, however, introduces additional computational complexity, particularly when PoIs are sparse. In such cases, larger isochrones are required to ensure that a PoI from every category is included, thus increasing the workload. Additionally, storing PoI coordinates also impacts RAM usage.

To overcome these limitations, we implemented a batch-processing approach that processes multiple nodes in parallel. However, the number of concurrent batches is constrained by the available RAM. As a result, larger numbers of nodes lead to longer computation times, while fewer PoIs also increase processing time. Conversely, an excessive number of PoIs reduces the number of parallel processes due to memory constraints.

In Table 2, we provide an overview of the cities with the highest number of nodes, edges, and PoIs to give an idea of the scale of our datasets and the computational demands of our approach.

To overcome the lack of computational power, we exploit the hardware and software resources provided by the Chameleon project [40]. The Chameleon project is a computing platform available for research purposes that provides computational resources and infrastructure for experimentation. The Chameleon's hardware includes nearly 15,000 cores, 5PB of total disk space, hosted across two sites, the University of Chicago and TACC, connected by 100 Gbps network. Moreover, Chameleon adopts OpenStack, a mainstream open source cloud technology, to provide its capabilities.

Here, we exploit the possibility provided by Chameleon to reserve bare metal machines which gives users full control over the machine's hardware and over the software stack including root privileges, kernel customization, and console access. Specifically, we ran experiments on the Chameleon testbed using an Intel(R) Xeon(R) Gold 6136 with 3.00GHz as processor frequency, 12 cores and 24 threads and 178GB of RAM.

Table 2. Summary of nodes, edges, POIs, and categories across different cities. Bold values indicate global max values.

|  | Tokyo Japan | Seoul South Korea | New York USA | Melbourne Australia | Shanghai China |
|---|---|---|---|---|---|
| Nodes | 624,593 | 156,887 | 380,525 | **762,917** | 172,548 |
| Edges | 1,817,772 | 451,574 | 1,259,360 | **2,253,012** | 488,444 |
| Total POIs | **232,273** | 154,297 | 142,197 | 80,536 | 133,334 |
| Mobility | 63,892 | 24,146 | 25,258 | 17,148 | **96,177** |
| Active Living | 66,021 | 16,314 | **91,431** | 48,497 | 17,310 |
| Entertainment | **12,161** | 8,390 | 2,675 | 1,474 | 986 |
| Food | 41,748 | **93,601** | 13,108 | 7,438 | 9,124 |
| Community | **9,440** | 2,015 | 1,047 | 745 | 624 |
| Education | **18,617** | 4,187 | 4,295 | 2,914 | 7,387 |
| Health & Wellbeing | **20,394** | 5,644 | 4,383 | 2,320 | 1,726 |

## Experimental analysis

### Analysis of accessibility metrics

We begin our analysis by ranking the 81 cities in our dataset (see the Data section for details on the selection process) by their *weighted average proximity*, similarly to previous studies [10,11,25]. In this ranking, each location's PoI-proximity is weighted by the number of residents, ensuring that densely populated areas contribute more significantly to the city's average proximity.

Fig 3 shows the distribution of these PoI-proximity values via box-plots, highlighting not only the differences across cities but also the variability within each city. We observe that 38 out of 81 cities have average weighted proximity values under 15 minutes, indicating the presence of relatively good walkable access to essential services for the majority of their residents. At the top of this ranking, we find cities like Paris with an average PoI-proximity of 5.3 minutes (IQR$\in [4.0, 6.0]$), Barcelona with 7 minutes (IQR$\in [4.0, 8.0]$), and Osaka with 9.1 minutes (IQR$\in [5.0, 9.0]$). Interestingly, cities at the top of the ranking (left-hand side of Fig 3) not only showcase lower average values but also have tighter inter-quartile ranges, meaning that they offer a more uniform access to services. The more we move down the ranking, however, the more the spread of the distributions increases. Cities with higher average PoI-proximity values, indeed, showcase wider distributions, revealing stronger spatial inequalities in access to services.

This is the case of cities at the bottom of the ranking (right-hand side of Fig 3), like Houston with an average PoI-proximity of 36.1 minutes (IQR$\in [26.0, 47.0]$), Jakarta with 34.6 minutes (IQR$\in [24.0, 45.0]$), and San Antonio with 34.4 minutes (IQR$\in [26.0, 43.0]$). The observed variability suggests that in lower-ranked cities, access to essential services is highly uneven, potentially exacerbating social and spatial inequalities within urban areas.

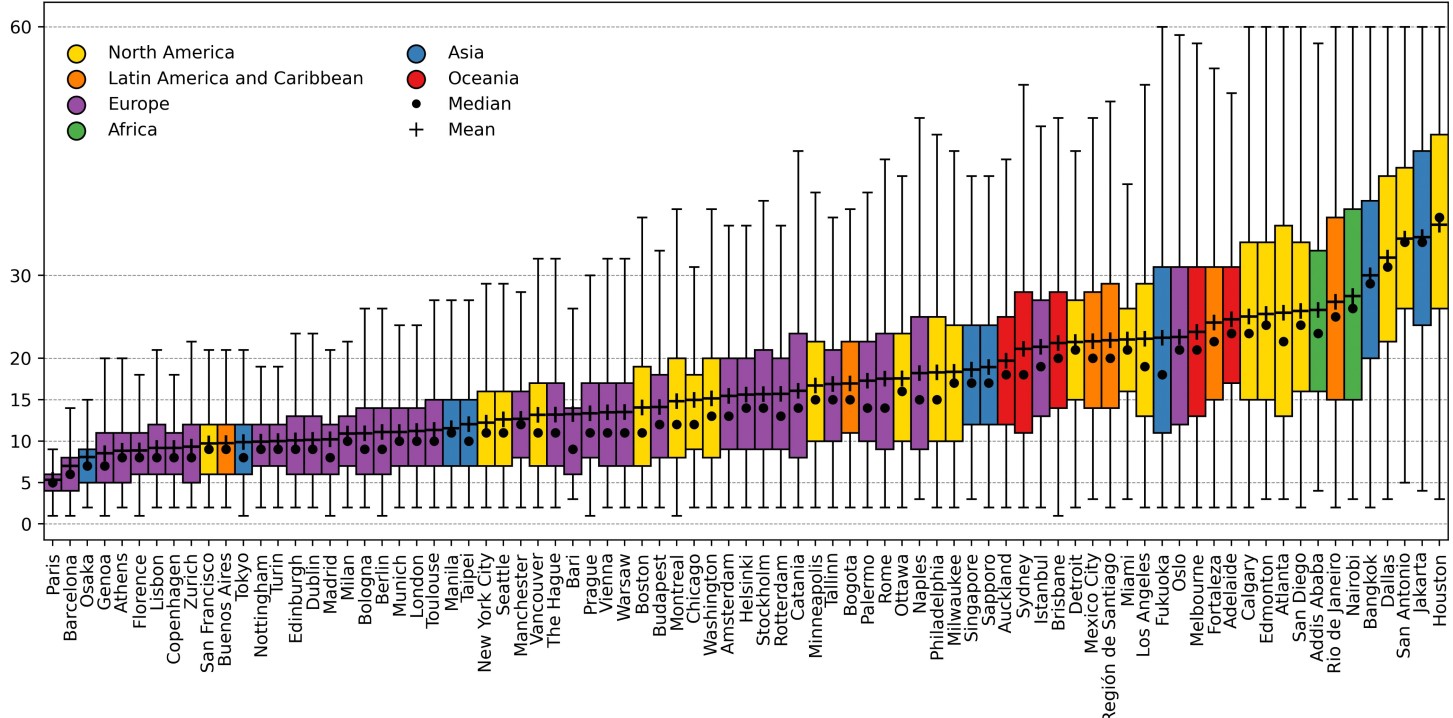

**Fig 3. Cities world-wide ranked by average PoI-proximity($\mathcal{P}$), weighted by population.** Each city is represented by a box-plot showing the distribution of $\mathcal{P}$ values across its area. Cities are sorted from lowest (left) to highest (right) weighted average proximity. Box-plots colors indicate the continent a city belongs to.

To better understand these aspects, we focus on a subset of representative cities for a more detailed comparison and fine-grained analysis. We extract these representative cities by computing a cumulative proximity curve, which describes the cumulative percentage of residents able to reach essential services within increasing walking time thresholds (i.e., those living in locations $n$ with PoI-proximity $\mathcal{P}(n)$).

To compare cities, we compute the area under the curve (AUC) as a summary indicator, where higher values indicate better accessibility — meaning more residents can reach essential services quickly — while lower values suggest poorer accessibility.

For the remainder of the analysis, we select six representative cities: the two cities with the highest and lowest AUC, plus one city at each of the 20th, 40th, 60th, and 80th percentiles. This selection includes Paris, Turin, Vancouver, Ottawa, Melbourne, and Houston, sorted by AUC, which we use in the rest of the analysis as reference points to illustrate differences in accessibility patterns. Their cumulative proximity curves are shown in Fig 4.

Fig 4 further characterizes the high heterogeneity between cities. We find that a staggering 99.7% of Paris citizens live in places with $\mathcal{P}(n) \leq 15$ minutes, compared to 90.5% for Turin, 75.0% for Vancouver, 58.8% for Ottawa, 40.1% for Melbourne, and only 9.8% for Houston. This progressive drop highlights a clear gradient in accessibility to services within walking distance. This result suggests that residents in lower-ranked cities may face greater challenges in meeting daily needs without relying on alternative means of transportation, potentially reinforcing car dependency and social inequalities.

However, PoI-proximity is just one dimension of accessibility. To capture a more comprehensive picture, we also compute additional indicators that account for the diversity and concentration of services available within a walkable distance. Specifically, for each node $n$ in our cities, we calculate PoI-density and PoI-entropy. Note that these measures rely on the isochrone $\mathcal{I}(n, t)$, which defines the reachable area within a given time. Here, we set the time parameter $t$ to the same

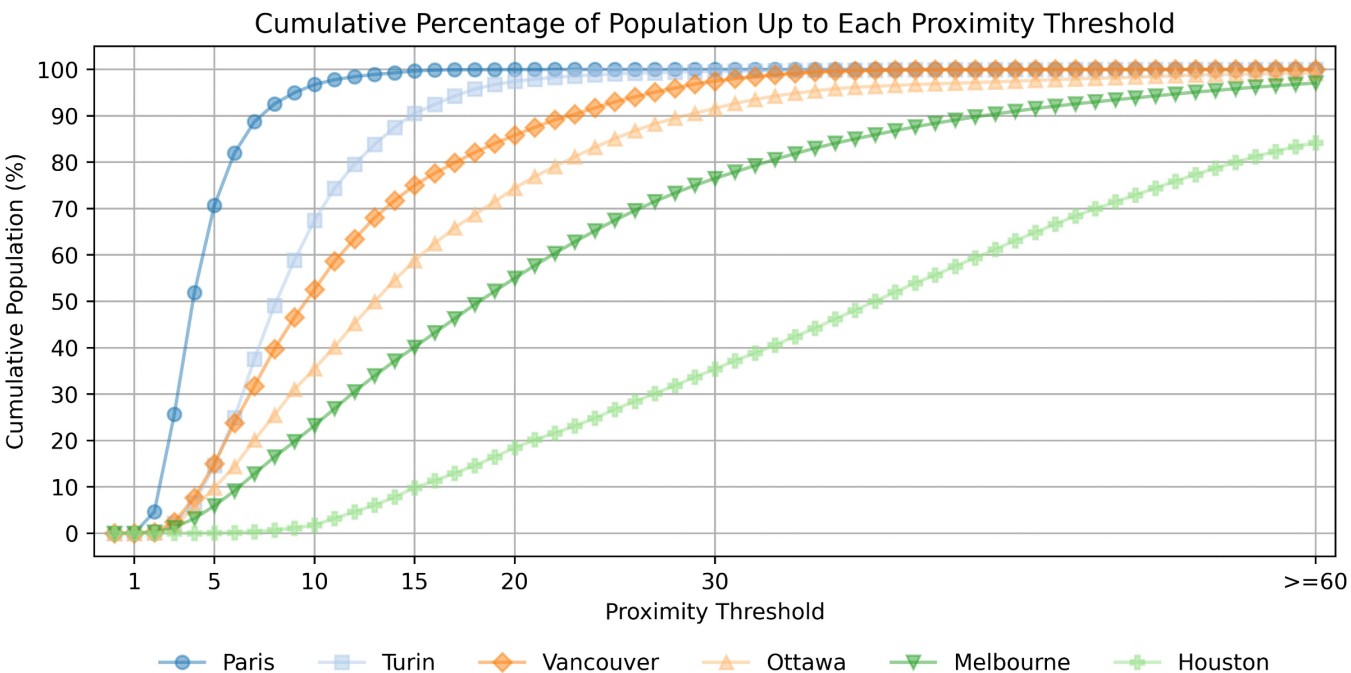

**Fig 4. Cumulative PoI-proximity curve for six representative cities.** Each curve shows the the percentage of the population within a given PoI-proximity threshold. The six cities include those with the highest and lowest area under the curve (AUC) values, plus one city at each of the 20th, 40th, 60th, and 80th percentiles.

value (e.g., *15* minutes) for all the isochrones so that we can fairly compare nodes, clusters, and ultimately cities between each other.

Fig 5 shows the distribution of these measures, alongside PoI-proximity, for the 6 cities on display, while other data can be found in the Supplementary Material (S3 Appendix: Distributions of $\mathcal{P}(n)$, $\mathcal{D}(15, n)$, $\mathcal{E}(15, n)$, and $\mathcal{A}(15, n)$ for all the other cities). Interestingly, when reading the figure from left to right, from Paris to Houston, we observe that the peak of the PoI-proximity $\mathcal{P}$ distribution — which indicates the number of city locations with services available within walkable distance — gradually shifts to the right and flattens. Analogously to the cumulative proximity curves, this trend reflects a progressive reduction in the number of areas with easily accessible services. A complementary trend is observed for $\mathcal{D}$, and $\mathcal{E}$: the peaks, initially located on the right side of the plots — indicating better density and entropy — progressively shift to the left, flattening the distribution profile as well. When focusing on our PoI-accessibility metric, $\mathcal{A}$, which combines PoI-proximity, PoI-density, and PoI-entropy (shown at the bottom of Fig 5), we observe broader distributions of values with fewer pronounced peaks (with the exception of Turin). However, the overall trend of the distributions shifting from higher accessibility values to lower values remains consistent. These results highlight a clear spatial disparity in access to services across cities, according to different measures. While some cities offer more uniformly distributed and accessible services, others exhibit more significant inequalities, with fewer areas having walkable access to essential services and greater variations in service diversity.

In principle, each of the metrics defined so far can be used independently to assess a specific dimension of city accessibility, as well as to rank and compare cities—or, within a city, different census areas, neighborhoods (i.e., nodes'

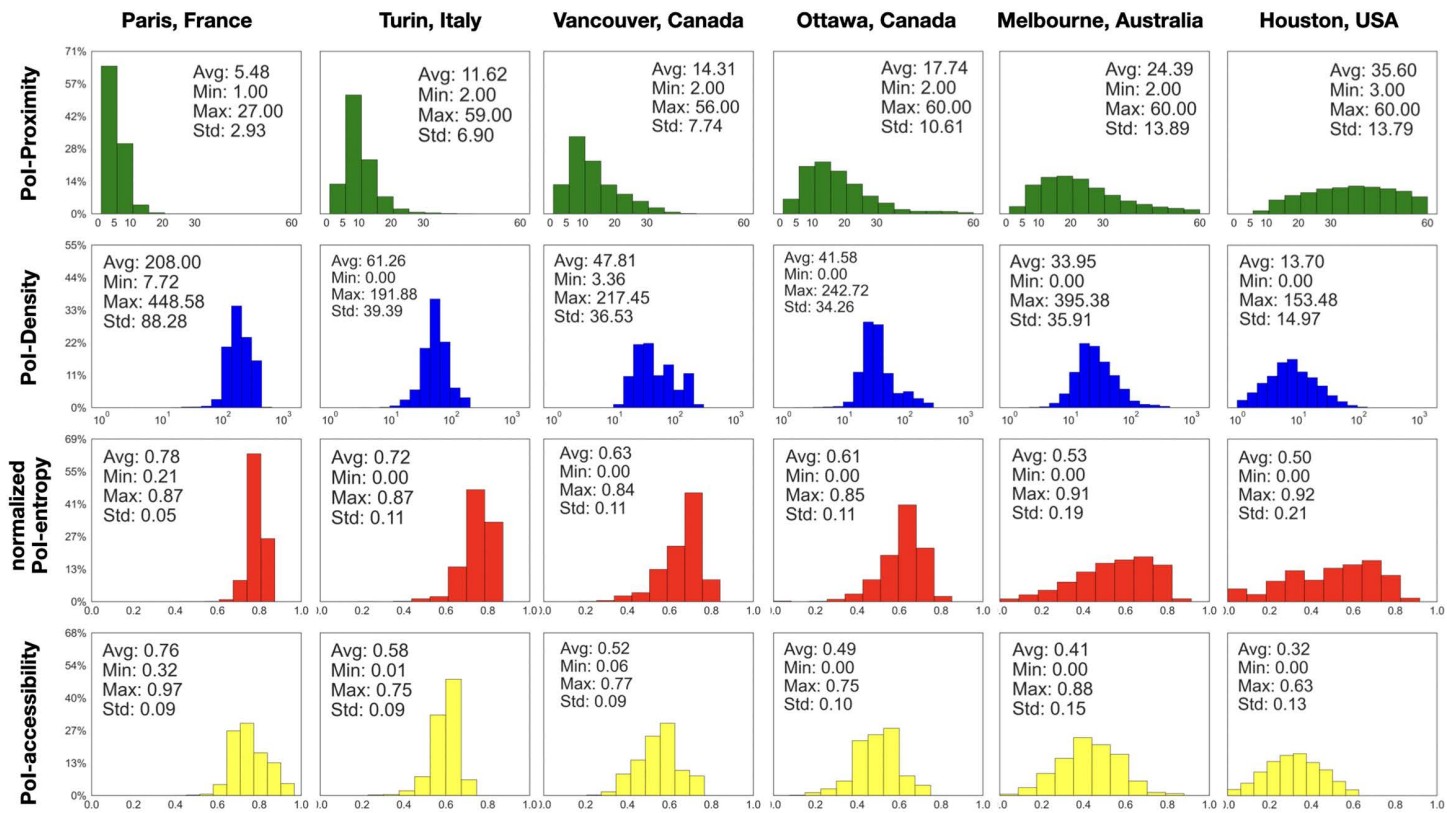

**Fig 5. Distributions of the accessibility metrics for six representative cities.** Shown are $\mathcal{P}(n)$ (PoI-proximity), $\mathcal{D}(15, n)$ (PoI-density within a 15-minute walk), $\mathcal{E}(15, n)$ (PoI-entropy within a 15-minute walk), and $\mathcal{A}(15, n)$ (PoI-accessibility within a 15-minute walk) across all nodes in each city.

clusters, which we use in this study), and so on. Fig 6 show how PoI-proximity, PoI-density, and PoI-entropy correlate with each other and with PoI-accessibility and closeness (that we will discuss later in more details), using both the Pearson coefficient and Kendall's tau. Correlation heatmaps reveal that all accessibility metrics are positively correlated with one another, while they either negatively correlate or show no significant correlation with both closeness and cities' population. Notably, PoI-proximity shows the highest correlation with PoI-accessibility, that, as defined in Eq. 1, is also a function of PoI-density and PoI-entropy. However, PoI-density and PoI-entropy also provide valuable perspectives: both of them show a strong correlation with PoI-proximity, but only a moderate correlation with each other, suggesting that they are likely to capture different aspects of accessibility. This highlights the importance of integrating all these viewpoints into a single measure, as achieved with PoI-accessibility.

This observation can be generalized by noting that rankings based on a single component (i.e., PoI-proximity, PoI-density, or PoI-entropy) are equivalent to rankings obtained by the PoI-accessibility measure when proper values are assigned to the coefficients $w_{\mathcal{P}}$, $w_{\mathcal{D}}$, and $w_{\mathcal{E}}$ in Eq. 1. For instance, setting $w_{\mathcal{P}} = 1$ and $w_{\mathcal{D}} = w_{\mathcal{E}} = 0$ reduces PoI-accessibility to PoI-proximity alone. More generally, adjusting these weights assigns more importance to certain components over the others, thus enabling more targeted analyses. For example, to highlight differences between urban areas primarily along the PoI-density dimension, one may use an unbalanced but non-exclusive weighting scheme, such as $w_{\mathcal{P}} = 1/2$, $w_{\mathcal{D}} = w_{\mathcal{E}} = 1/4$. Interestingly, such alternative weightings do not substantially affect the overall rankings, as correlations remain strong or very strong (see the Supplementary Material S2 Appendix: More on cities' rankings for more details).

To provide a more comprehensive comparison, we rank all cities based on their PoI-accessibility, as shown in Fig 7(a). Similar to the previous analysis, we use a box-plot per city, with average PoI-accessibility weighted by the number of

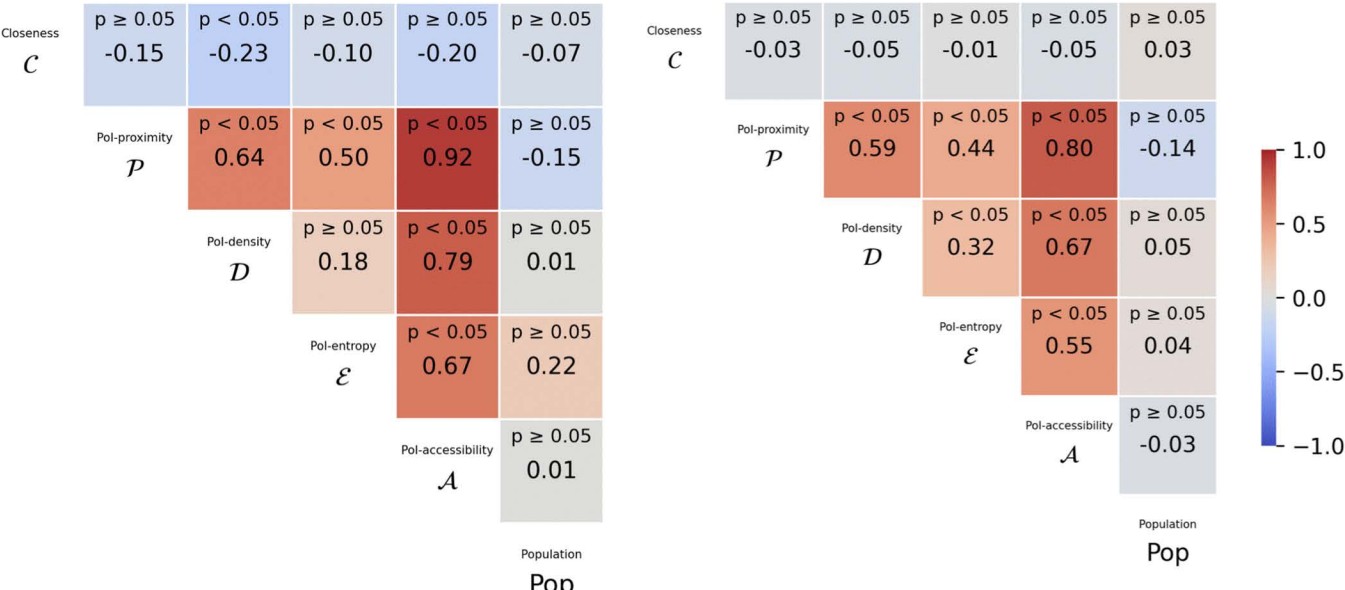

(a) Pearson coefficient heatmap          (b) Kendall's tau heatmap

**Fig 6. Correlation heatmaps for key urban metrics across 81 cities.** The matrices show pairwise correlations between PoI-proximity, PoI-density, PoI-entropy, PoI-accessibility, closeness, and population using **(a)** Pearson correlation coefficients and **(b)** Kendall's tau. Colors range from −1 (blue, strong negative correlation) to 1 (red, strong positive correlation), with gray indicating near-zero correlation. Reported values indicate the corresponding correlations and p-values.

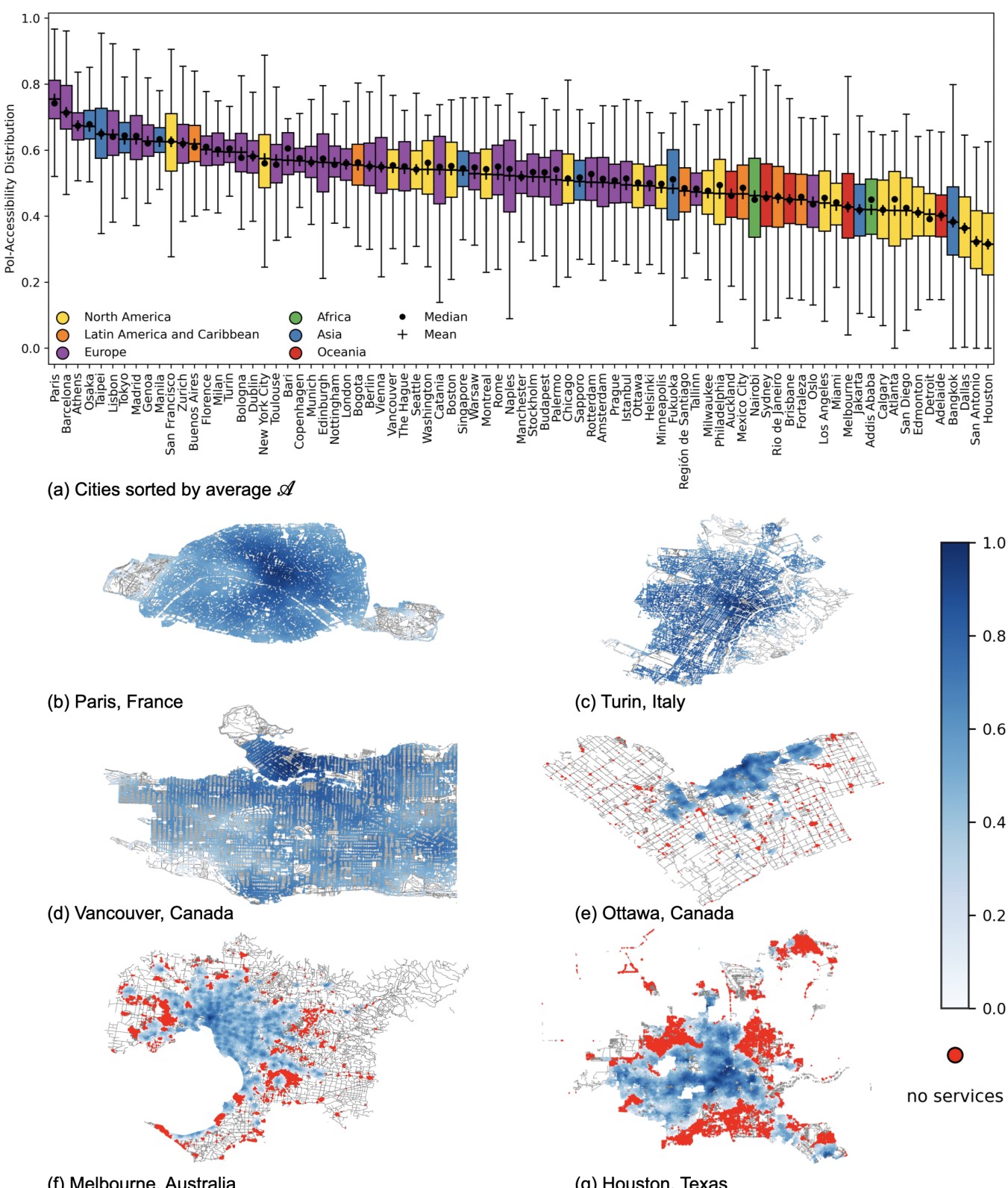

(a) Cities sorted by average 𝒜

(b) Paris, France

(c) Turin, Italy

(d) Vancouver, Canada

(e) Ottawa, Canada

(f) Melbourne, Australia

(g) Houston, Texas

**Fig 7. PoI-accessibility ranking and its spatial patterns for the six representative cities. (a)** Cities world-wide sorted in descending order of average PoI-accessibility 𝒜 weighted by population, with $t = 15$. **(b-g)** Heatmaps of 𝒜 values for the six representative cities at the intersection level. Points

heatmaps are color coded by population-weighted $\mathcal{A}$ (scale shown on the right): darker areas indicate better access to services, white areas have no (or a negligible number of) residents, and red areas represent nodes with no access to services. *Maps contain information from OpenStreetMap and OpenStreetMap Foundation, which is made available under the Open Database License.*

residents in each location. While there is a high statistically significant correlation (Kendall's Tau of 0.8 with $p < 0.05$, as in Fig 6(b)) between the PoI-proximity and PoI-accessibility, we can already observe some key differences between the rankings based on them. These differences impact positions across the entire ranking, including both the top and bottom. For example, Taipei, which was previously ranked 25th in terms of PoI-proximity, now ranks in the top 5 by accessibility. Similarly, Jakarta, which was second to last in PoI-proximity, has gained 10 positions, significantly improving its ranking. These shifts showcase that, even though proximity and accessibility rankings are strongly correlated, our accessibility metric offers a more nuanced perspective by incorporating the additional dimensions of entropy and density.

This aspect is supported by the spatial distribution of cities' accessibility. By visualizing our PoI-accessibility metric through heatmaps (see Fig 7), we observe that in Paris, areas with high accessibility (darker colors) are more homogeneously distributed across the city. In contrast, as we move towards cities with lower proximity AUC values, accessibility becomes increasingly heterogeneous, with well-served areas becoming more scattered throughout the city. Comparing areas within cities, the heatmaps in Fig 7 further reveal a general core-periphery pattern: PoI-accessibility tends to decrease as we move from the city center towards peripheries. However, not all the peripheral areas are alike. Some have no or very few residents, resulting in PoI-accessibility values close to 0, while others contain nodes where PoI-proximity is greater than 60 minutes. These nodes are highlighted in red in Fig 7 and labeled as "no services".

These observations help clarify why, despite the high correlation between $\mathcal{P}$ and $\mathcal{A}$, relying solely on PoI-proximity would be a misleading simplification. For instance, an area with a single shopping mall may exhibit high PoI-proximity, as access to services is quick, but these services are concentrated in one location (low PoI-density). Furthermore, the shopping mall may primarily contain services within the food and entertainment categories, while the health and well-being category may be represented solely by the presence of a single pharmacy (low PoI-diversity).

We illustrate this scenario in Fig 8 (left), showing an intersection $n_x$ in Paris with a good PoI-proximity score (i.e., $\mathcal{P}(n_x) = 5$ minutes), but with a relatively poor PoI-entropy ($\mathcal{E}'(n_x) = 0.73$, below the 2th percentile in Paris) and poor PoI-density ($\mathcal{D}'(n_x) = 0.20$, below the median in Paris). The 15-minute isochrone $\mathcal{I}(n_x, 15)$ contains a significant number of "active living" PoIs (primarily bicycle parkings) and "mobility" PoIs (public transport stops), which contribute to increase PoI-density but also to keep PoI-entropy low. While there are several "food-related" PoIs, the closest marketplace is located 12 minutes away. Similarly, in the "health and well-being" category, multiple pharmacies are present nearby, but no hospitals, and the nearest doctor is at the edge of the 15-minute isochrone. On the right of Fig 8, instead, we present an example of an intersection $n_y$ with poor PoI-proximity ($\mathcal{P}(n_y) = 15$ minutes, above the median), but good PoI-entropy and PoI-density ($\mathcal{E}'(n_x) = 0.88$, and $\mathcal{D}'(n_x) = 0.56$, both above the median). In this case, proximity is poor because the closest service from the "community" category is available only at a 15-minute distance, while all the other categories are dense and well-distributed within even smaller isochrones. These examples make clear that PoI-proximity alone is not enough to not guarantee diverse or balanced access to services.

## Analysis of closeness and topological isolation

While accessibility provides a snapshot of how well-served a specific location is, it does not reveal how connected that location is to the rest of the city. This is where closeness comes into play. As described in the Methods section, we calculated the normalized closeness $\mathcal{C}'(n)$ for each point $n$ of a city. Closeness estimates how residents in one area can easily reach other parts of the city, with the assumption that residents in $n$ can walk or make use of the urban transport system to get to the target. Thus, closeness captures aspects of urban connectivity and topological isolation that accessibility alone

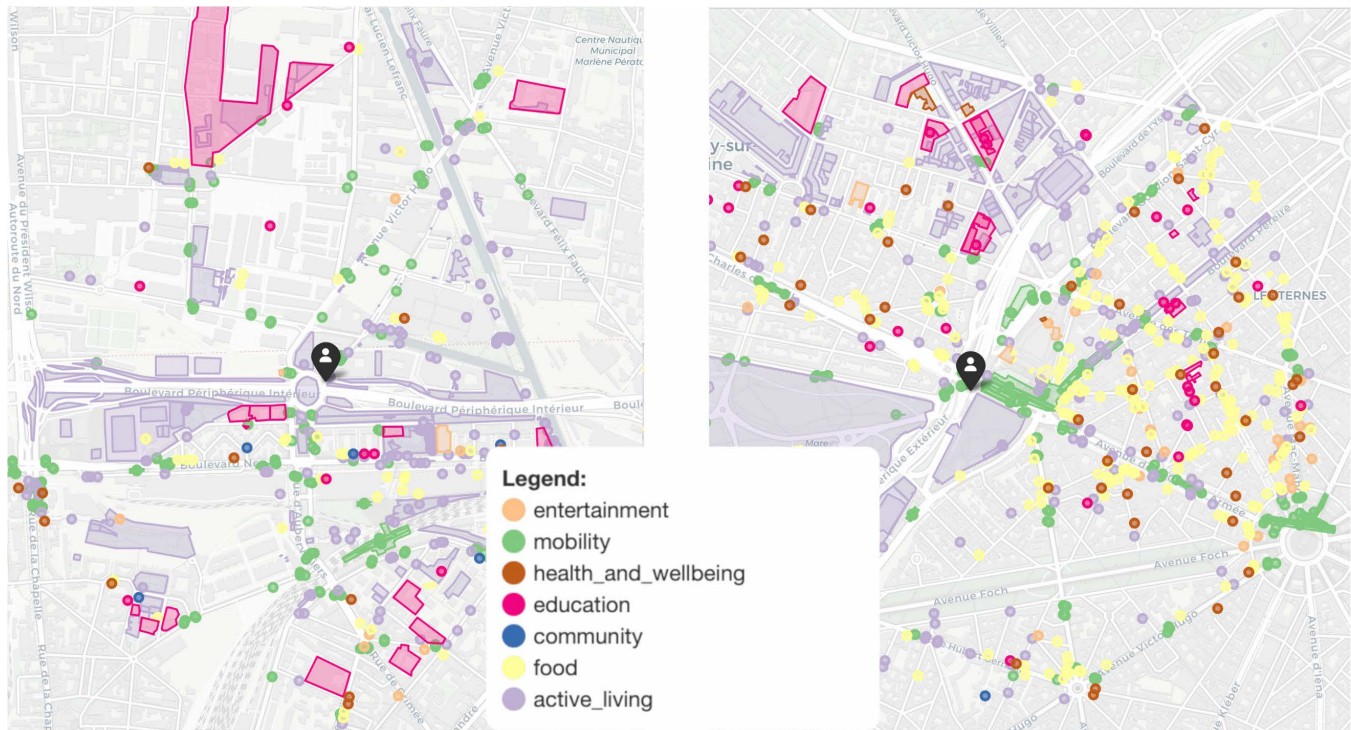

**Fig 8. Comparison between PoIs within 15-minute isochrones for two example intersections in Paris.** (Left) An intersection with good PoI-proximity but poor PoI-entropy, and PoI-density. (Right) An intersection with poor PoI-proximity but good PoI-entropy and PoI-density. Colored areas refer to the presence of PoIs in a certain category. *Base map and data from OpenStreetMap and OpenStreetMap Foundation*.

does not. In the following, we use the normalized closeness (see the Methods section for details on its definition and computation) not only to compare cities but also to compare different parts within cities.

First, we rank cities by their average closeness values $\mathcal{C}'(n)$, where values are weighted by the fraction of the population in $n$. This ranking, shown in Fig 9(a), presents a quite different picture compared to the PoI-accessibility ranking. Looking at specific examples, we observe notable changes in city ranking when switching from accessibility to closeness. Focusing on the top 5 by PoI-accessibility, for example, we have that Paris, which ranks 1st in PoI-accessibility, drops to 71st in terms of closeness. Similarly, Osaka moves from 4th position to 79th, and Taipei drops from 5th position to 77th. On the other hand, cities like Barcelona and Athens, which ranked 2nd and 3rd by PoI-accessibility, shift to 16th and 38th respectively in terms of closeness. These shifts highlight how the closeness measure, which reflects the connectivity between areas within a city, provides a different perspective, revealing more potentially segregated urban patterns compared to PoI-accessibility. This is further supported by the statistically non-significant Kendall's Tau correlation value of $-0.04$ ($p \geq 0.05$) between the PoI-accessibility and closeness metrics (see Fig 6(b)).

To better understand the heterogeneity of cities in terms of closeness, we examine the spatial distribution of its values through heatmaps. In Fig 9, we show the six representative cities' heatmaps describing how $\mathcal{C}'$ is distributed in the urban landscape. Not surprisingly, the best connected points in the urban transport networks of cities usually coincide with the topological center, especially for European cities (e.g., in Paris, closeness is maximum at Musée d'Orsay — with no reported residents around). Still, we can easily spot faster backbones, as well as well-connected hubs scattered within a city, making some neighborhoods (and some peripheries) better connected than others. This is evident especially for Vancouver, which also ranks 5th by closeness. Overall, these patterns demonstrate how urban connectivity is not uniformly

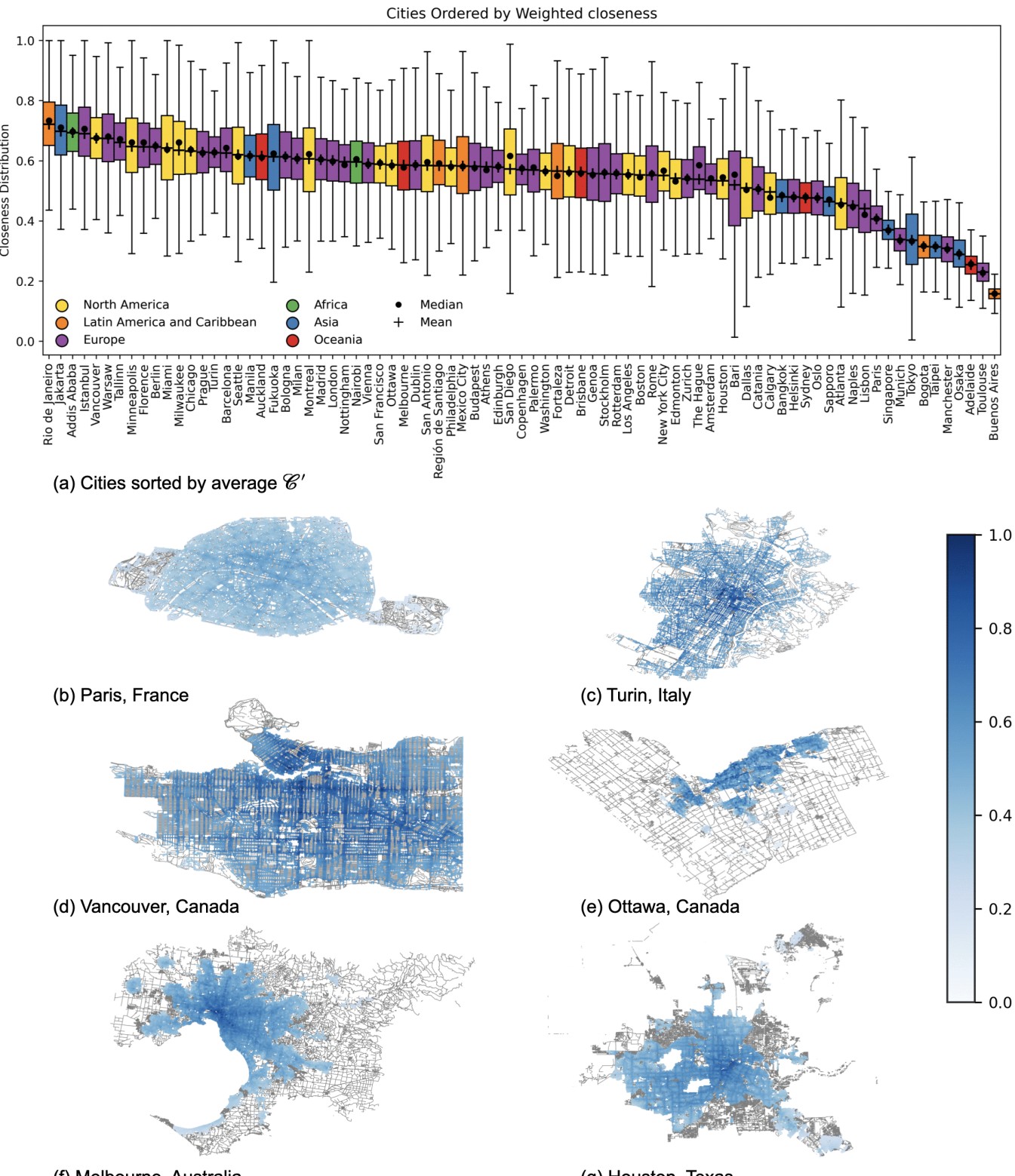

**Fig 9. Closeness ranking and its spatial patterns for the six representative cities. (a)** Cities world-wide sorted in descending order of average closeness $\mathcal{C}'$, weighted by population. **(b-g)** Heatmaps of $\mathcal{C}'$ values for the six representative cities at the intersection level. Points are color coded by

population-weighted $\mathcal{C}'$ (scale shown on the right): darker areas indicate higher closeness, while white areas correspond to nodes with no (or a negligible number of) residents, no connection to the city network, or both. *Maps contain information from OpenStreetMap and OpenStreetMap Foundation, which is made available under the Open Database License.*

distributed, with some areas enjoying better integration into the city's transport network, while others remain more isolated despite their accessibility to services.

### Comparing closeness and PoI-accessibility at different scales

In the following, we delve deeper into the interplay between PoI-accessibility and normalized closeness, to understand how inequalities manifest at different scales, shedding lights on potential urban segregation phenomena that can be overlooked by accessibility measures alone.

We first compare PoI-accessibility and normalized closeness at a coarse granularity by plotting the 81 cities in our list in a single bubble chart. As shown in Fig 10, there is no clear relationship between the two metrics. This is further supported by the Pearson coefficient ($\approx -0.2$, p-value $\approx 0.07$), and Kendall's Tau ($\approx -0.05$, p-value $\approx 0.5$ – see Fig 17) which are not statistically significant. However, we can distinguish two groups of cities: some cities (such as Adelaide, Barcelona, Bogota, Buenos Aires, and Paris) are scattered above the diagonal, while others tend to fall below or along it (such as Houston, Jakarta, Melbourne, Ottawa, Rio de Janeiro, Turin, and Vancouver). To visually capture these differences, we ran an elliptic envelope outliers detection system [41] (with contamination parameter set to 0.22), and marked the outliers in the figure with a different texture. For a comparison between PoI-accessibility and closeness at a geographical regions resolutions, the reader can find other bubble charts in the Supplementary Material (S4 Appendix: Cities' bubble charts by geographical region). Also, to compare each of the three components of PoI-accessibility, namely PoI-proximity, PoI-density, and PoI-entropy, to closeness, we plotted three more bubble charts that can be found in the Supplementary Material (S7 Appendix: Comparing Closeness to $\mathcal{P}$, $\mathcal{D}$, and $\mathcal{E}$).

To sum up, we find that some cities have adequate/high accessibility, but low closeness (e.g., Buenos Aires, Bogota, Paris), while other cities are at the opposite side of the spectrum with low accessibility and adequate/high closeness (e.g., Rio de Janeiro, Jakarta), and cities that under perform (e.g., Adelaide, Houston) or over perform (e.g., Barcelona, Turin) in both measures. PoI-accessibility and normalized closeness medians are also drawn in the plot to identify the above mentioned four regions.

To gain a finer-grained understanding of how accessibility and connectivity interact within each city, we next shift our focus to the cluster level. For each city, we detect clusters using Infomap and compare these "natural" neighborhoods in terms of both accessibility and closeness, as shown in Fig 11, and in Fig 12. While the relationship between the two metrics remains imperfect, it seems to be stronger at this scale: areas with higher accessibility tend to also show better closeness. This is reflected in stronger correlation values within cities — from 0.62 for Ottawa (where cluster 7 stands out with high closeness but low accessibility) to 0.98 for Melbourne (see Table 3).

Yet, even in cities ranking high for accessibility, disparities in closeness remain evident. Paris, for example, shows consistently high accessibility scores across all clusters, but most neighborhoods still display low closeness values, between 0.35 and 0.45 — with the exception of cluster 6, which has an average of 0.48. This suggests that meeting the 15-minute city goal does not automatically ensure strong integration within the transport network.

At an even more fine-grained resolution, we relate PoI-accessibility to normalized closeness at the level of individual intersections, as in Fig 11, and in Fig 12. While these scatter plots are denser and more complex to interpret, they further showcase the variability within cities. Even cities with strong overall accessibility (e.g., Paris) or high connectivity (e.g., Turin) exhibit significant internal heterogeneity, showing that good city-level scores can still mask local disparities across neighborhoods. The reader can find the above mentioned plots at a greater resolution in the Supplementary Material (S5 Appendix: Plots on the 5 representative cities at a greater resolution and S6 Appendix: More details on Italian cities).

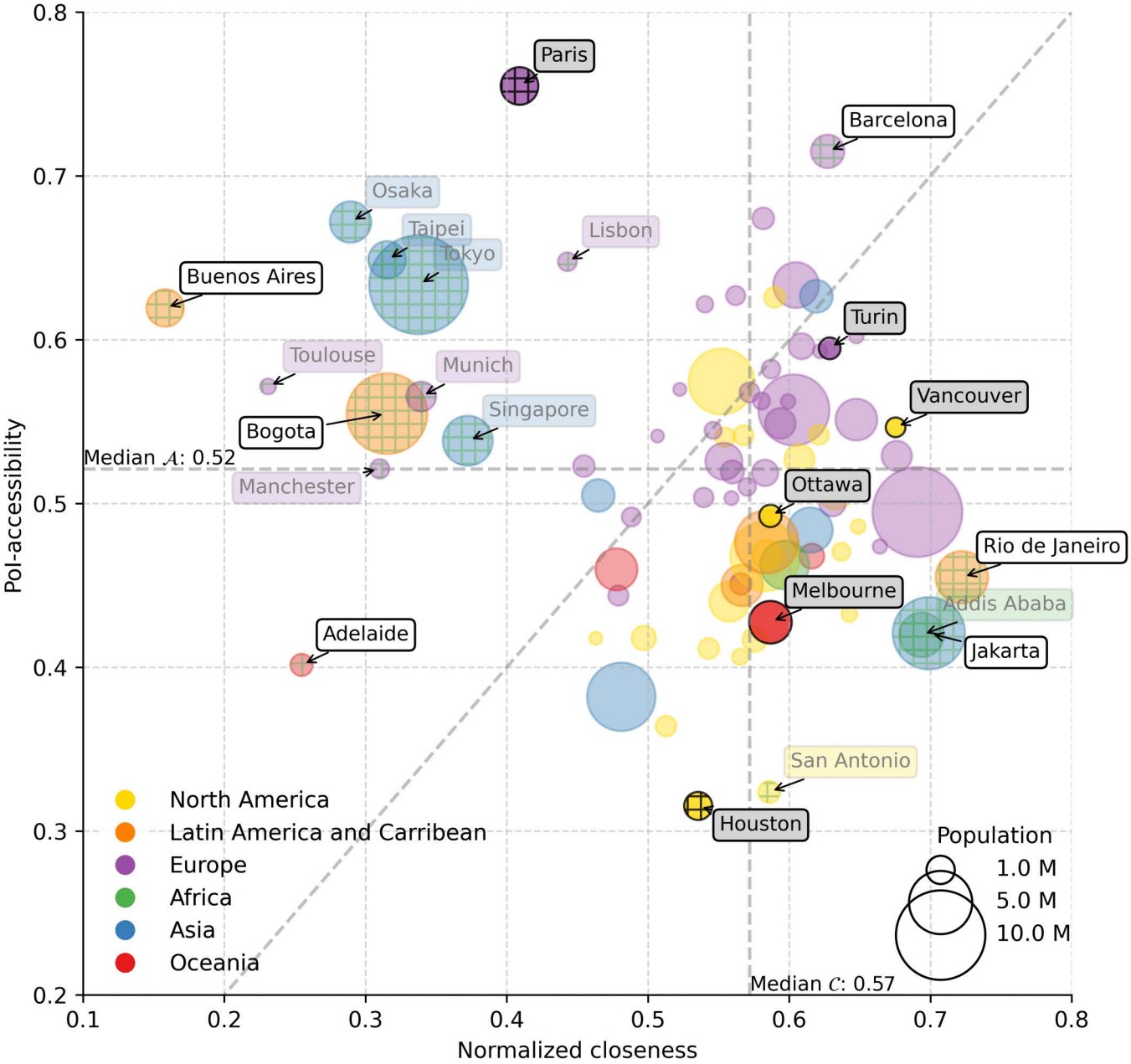

**Fig 10. Comparison of Pol-accessibility and normalized closeness across cities.** Bubble chart showing cities' Pol-accessibility and normalized closeness. Marker size is proportional to city population, and marker color refers to the geographical region a city belongs to. Outliers, identified using an elliptic envelope (with contamination parameter of 0.22), are textured. Text annotations' backgrounds are color-coded by geographical region, except for the six representative cities (light gray) and outliers discussed in the text (white).

Our multiscale analysis reveals a wide heterogeneity at every level. Some cities are both well-connected and highly accessible, while others have high accessibility without necessarily ensuring good internal connectivity. Within cities, we observe significant variability: although the correlation between the two main dimensions is generally strong, in cities like Paris and Turin, some clusters with a non-negligible number of residents still experience poorer proximity to services and

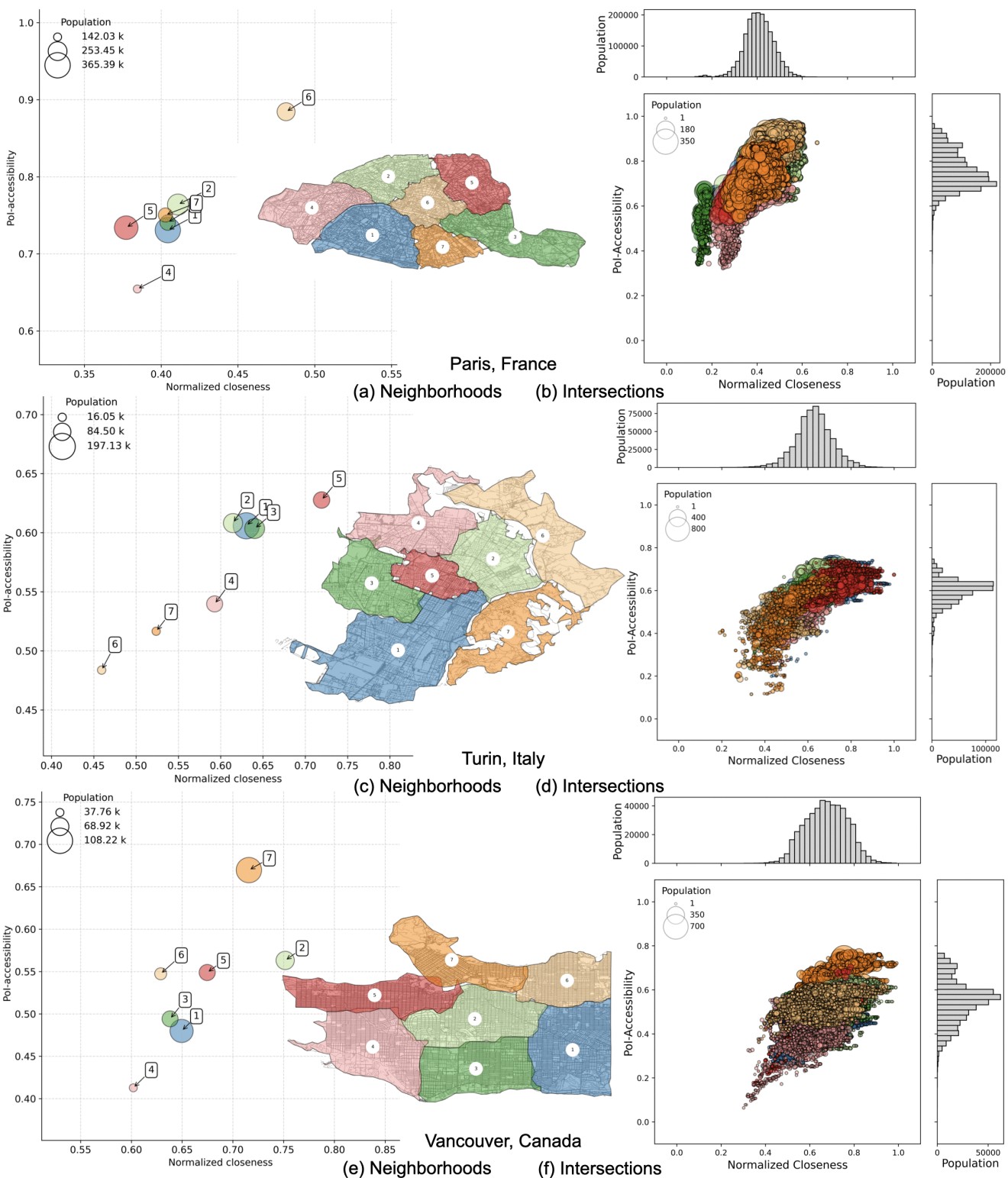

**Fig 11. Pol-accessibility vs. normalized closeness for Paris (a, b), Turin (c, d) and Vancouver (e, f).** Left panels: each bubble corresponds to a neighborhood detected by the Infomap algorithm and color-coded accordingly. Axes are ranged differently, to simplify the comparison between neighborhoods within a city. Right panels: nodes represent intersections, with sizes proportional to the number of residents. Population distributions are shown as histograms along the axes. *Maps contain information from OpenStreetMap and OpenStreetMap Foundation, which is made available under the Open Database License.*

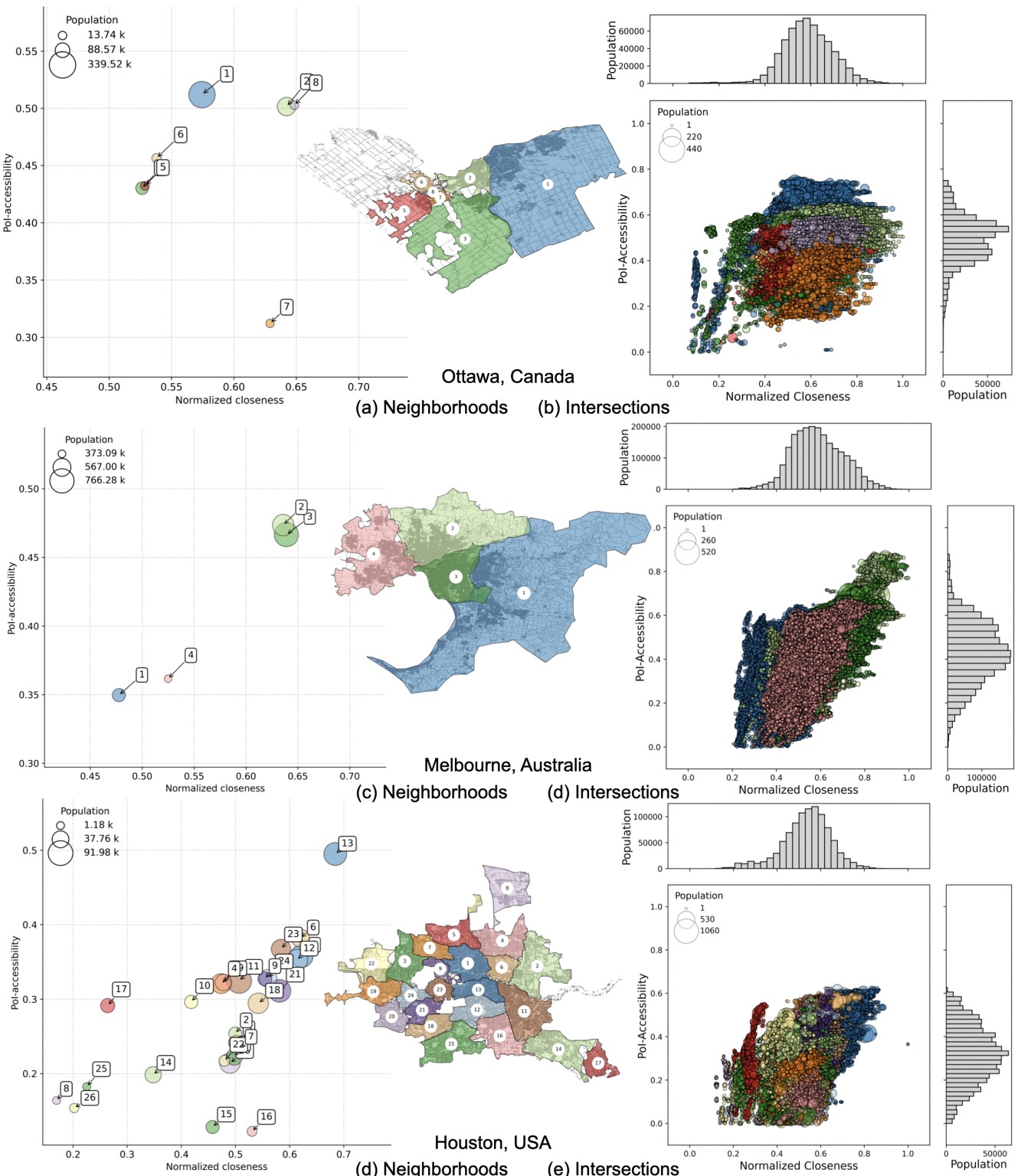

**Fig 12. Pol-accessibility vs normalized closeness for Ottawa (a, b), Melbourne (c, d), and Houston (e, f).** Left panels: each bubble corresponds to a neighborhood detected by the Infomap algorithm and color-coded accordingly. Axes are ranged differently, to simplify the comparison between neighborhoods within a city. Right panels: nodes represent intersections, with sizes proportional to the number of residents. Population distributions are shown as histograms along the axes. *Maps contain information from OpenStreetMap and OpenStreetMap Foundation, which is made available under the Open Database License.*

**Table 3. Summary statistics of the six representative cities. Mean and the standard deviation of the PoI-accessibility and normalized closeness are aggregated by Infomap clusters. The Pearson r and Kendall's τ coefficients are calculated between the two variables.**

| City | $\langle A \rangle$ | $\sigma(A)$ | $\langle C' \rangle$ | $\sigma(C')$ | $r$ | $\tau$ |
|---|---|---|---|---|---|---|
| Paris | 0.75 | 0.06 | 0.41 | 0.03 | 0.91 | 0.52 |
| Turin | 0.57 | 0.05 | 0.60 | 0.08 | 0.94 | 0.71 |
| Vancouver | 0.53 | 0.07 | 0.67 | 0.05 | 0.74 | 0.62 |
| Ottawa | 0.45 | 0.06 | 0.51 | 0.05 | 0.62 | 0.43 |
| Melbourne | 0.41 | 0.06 | 0.41 | 0.07 | 0.98 | 0.67 |
| Houston | 0.27 | 0.09 | 0.48 | 0.13 | 0.66 | 0.60 |

weaker connectivity to the rest of the city. At the neighborhood level, we can see that even single clusters show striking inequalities in accessibility and closeness distributions.

Such diversities may result from a combination of historical, geographical, economic, and social factors. Though investigating the underlying causes of livability and urban segregation is beyond the scope of this study, we can start drawing some conclusions. While the hyper-proximity concept promotes good practices and metrics to assess and design more livable neighborhoods, our analysis does not provide empirical evidence that cities performing better according to this model are also becoming more inclusive or effectively overcoming urban segregation.

### Socio-demographic analysis: Italian cities case study

To explore potential links between accessibility, connectivity, and socio-economic conditions, we perform a simple comparison between the average income of Italian cities, PoI-accessibility, and normalized closeness. We aim at identifying possible signals of correlation between low accessibility or poor network integration and lower residents' income, as a proxy for segregation.

We perform this analysis on the ten most populous Italian cities (i.e., Bari, Bologna, Catania, Florence, Genoa, Milan, Naples, Palermo, Rome, and Turin). However, for the sake of brevity, we exclude Catania in this section an focus on the biggest nine Italian cities. We provide additional results, including those for Catania, in the Supplementary Material (S6 Appendix: More details on Italian cities). In Fig 13, we show two bubble charts comparing PoI-accessibility and normalized closeness across Italian cities, enriched with additional information on population size (left panel), average income (right panel), and geographical macro-regions (Northern, Central, and Southern Italy).

Looking at population, we observe no clear relationship between city size and our metrics. Both large cities, such as Milan and Turin, and smaller ones, like Bologna and Florence, have higher accessibility and closeness values. Conversely, some large cities like Rome perform poorly, similar to smaller southern cities like Bari.

The more interesting pattern emerges when looking at average income. Here, a clearer trend emerges: cities with higher average income tend to perform better on both accessibility and normalized closeness, reflecting the well-known North-South divide in Italy, with the exception of Florence, from Central Italy, which aligns more closely with the northern group. These results suggest that economic disparities are not only visible in income but also mirrored in the urban structure and service distribution.

In Table 4, we show basic statistics about PoI-accessibility and normalized closeness within these cities, stressing that the two measures correlate well at this level: Kendall's tau returns very strong (Bari) or strong (Genoa, Florence, Milan, Bologna, Turin, Rome) signals, meaning that isolation is more likely in neighborhoods that are also poorly served. Naples and Palermo show a moderate ranking correlation signal.

We also plot neighborhoods' multi dimensional comparisons for Bari, Bologna, Florence (Fig 14), Genoa, Milan, Naples (Fig 15), Palermo, Rome, and Turin (Fig 16). Interestingly, we find districts that show a signal of voluntary isolation (like

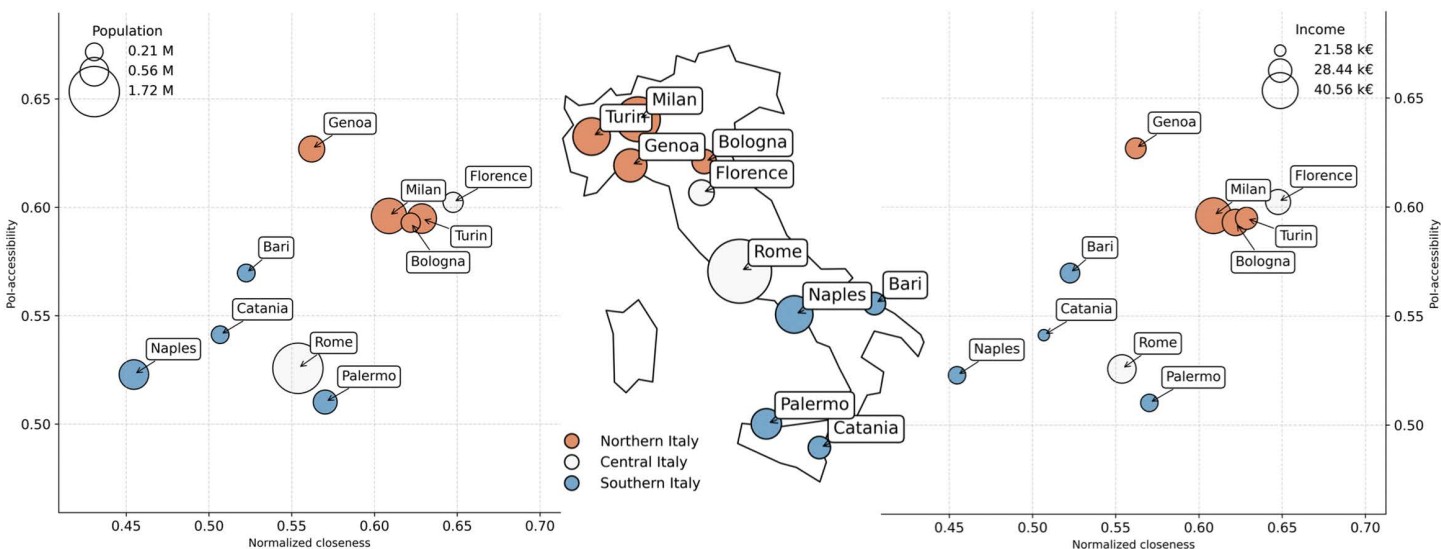

**Fig 13. Pol-accessibility vs. normalized closeness for the nine most populous Italian cities.** Two bubble charts comparing cities by Pol-accessibility (y-axis) and normalized closeness (x-axis) with marker size proportional to city population (left) and to average income (right). Markers are color-coded according to three geographical macro-regions (North, Center, South). A central map of Italy displays the location of the nine cities, with sizes proportional to city population, providing geographical context for the comparison.

**Table 4. Summary statistics of the top nine Italian cities by population.** The mean and standard deviation of the Pol-accessibility, normalized closeness, and the average income are aggregated by Infomap clusters. The Pearson *r* and Kendall's *τ* coefficients are calculated between the first two variables.

| City | $\langle A \rangle$ | std($A$) | $\langle C \rangle$ | std($C$) | *r* | *τ* | avg income | std income |
|---|---|---|---|---|---|---|---|---|
| Genoa | 0.624 | 0.058 | 0.561 | 0.084 | 0.925 | 0.810 | 26,254.124 | 4,982.607 |
| Florence | 0.596 | 0.051 | 0.641 | 0.054 | 0.947 | 0.867 | 28,885.842 | 3,116.686 |
| Milan | 0.585 | 0.045 | 0.592 | 0.064 | 0.935 | 0.889 | 36,370.690 | 10,275.040 |
| Bologna | 0.579 | 0.056 | 0.605 | 0.066 | 0.884 | 0.733 | 28,680.824 | 3,787.589 |
| Turin | 0.569 | 0.051 | 0.597 | 0.078 | 0.939 | 0.714 | 29,608.103 | 9,854.723 |
| Bari | 0.553 | 0.084 | 0.496 | 0.123 | 0.967 | 1.000 | 24,624.574 | 3,225.174 |
| Naples | 0.505 | 0.081 | 0.454 | 0.074 | 0.877 | 0.600 | 20,736.677 | 4,162.221 |
| Rome | 0.495 | 0.079 | 0.505 | 0.114 | 0.900 | 0.795 | 29,985.519 | 7,110.777 |
| Palermo | 0.472 | 0.119 | 0.535 | 0.084 | 0.763 | 0.613 | 21,350.120 | 4,017.969 |

in the so-called gated communities that are usually reported in American or Asian cities). In fact, some neighborhoods with average income higher than the city's average, also have very low $\mathcal{A}$ and $\mathcal{C}'$ values. This phenomenon is particularly evident for cluster 2 in Bologna (Fig 14); cluster 2 in Genoa, and cluster 1 in Naples (Fig 15); cluster 17 in Palermo, cluster 1 in Rome, and cluster 7 in Turin (Fig 16). This scenario suggests that the wealthiest residents may be even more unlikely to quit relying on cars than expected in the near future, weakening the application of the 15-minute city even in urban landscapes where such a model is apparently already in place. These observations make a city like Milan (Fig 15) an interesting exception: the correlation between Pol-accessibility and normalized closeness is very strong (Table 4), and although residents in the less served neighborhoods (i.e., clusters 3 and 6) are characterized by an average income lower than the city average, it should be noted that their Pol-accessibility and normalized closeness are very close to the global medians shown in Fig 10.

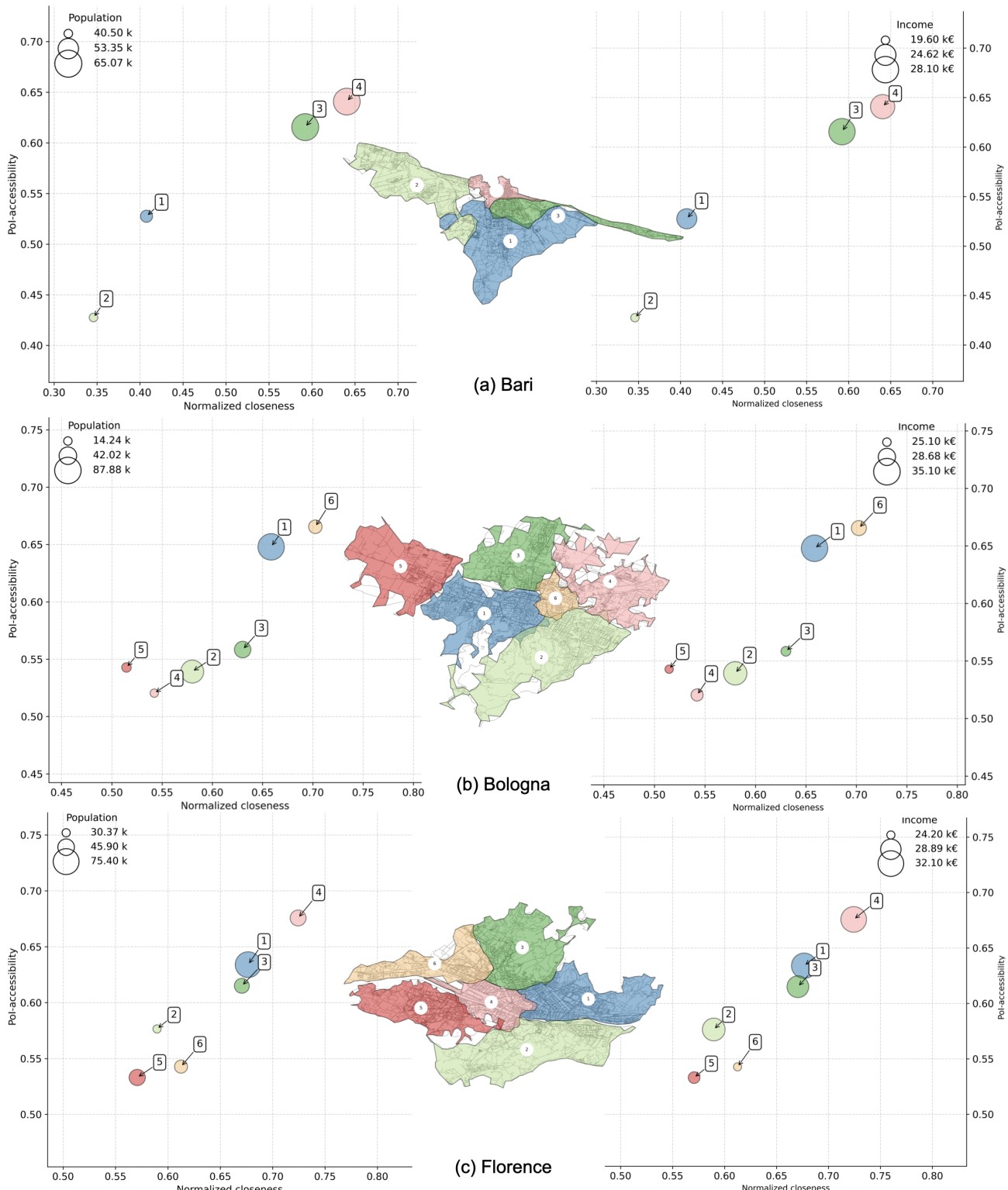

**Fig 14. PoI-accessibility vs. closeness by neighborhoods in Bari (a), Bologna (b), and Florence (c).** Markers are color-coded according to the administrative neighborhoods (as shown in the embedded city map), and sized by population (left) and average income (right). Axes are ranged differently, to simplify the comparison between neighborhoods within a city. *Maps contain information from OpenStreetMap and OpenStreetMap Foundation, which is made available under the Open Database License.*

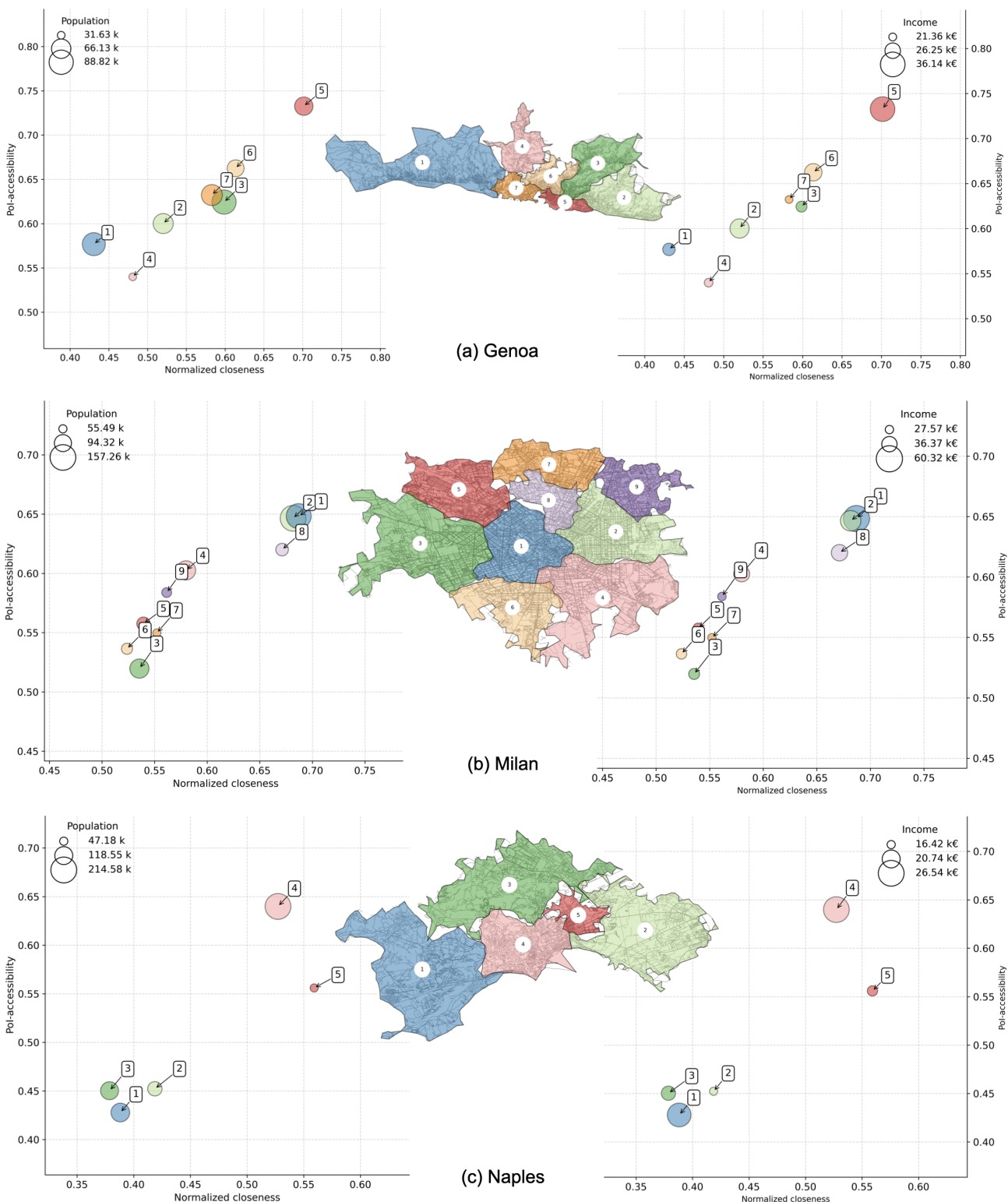

**Fig 15. PoI-accessibility vs closeness by neighborhoods in Genoa (a), Milan (b), and Naples (c).** Markers are color-coded according to the administrative neighborhoods (as shown in the embedded city map), and sized by population (left) and average income (right). Axes are ranged differently, to simplify the comparison between neighborhoods within a city. *Maps contain information from OpenStreetMap and OpenStreetMap Foundation, which is made available under the Open Database License.*

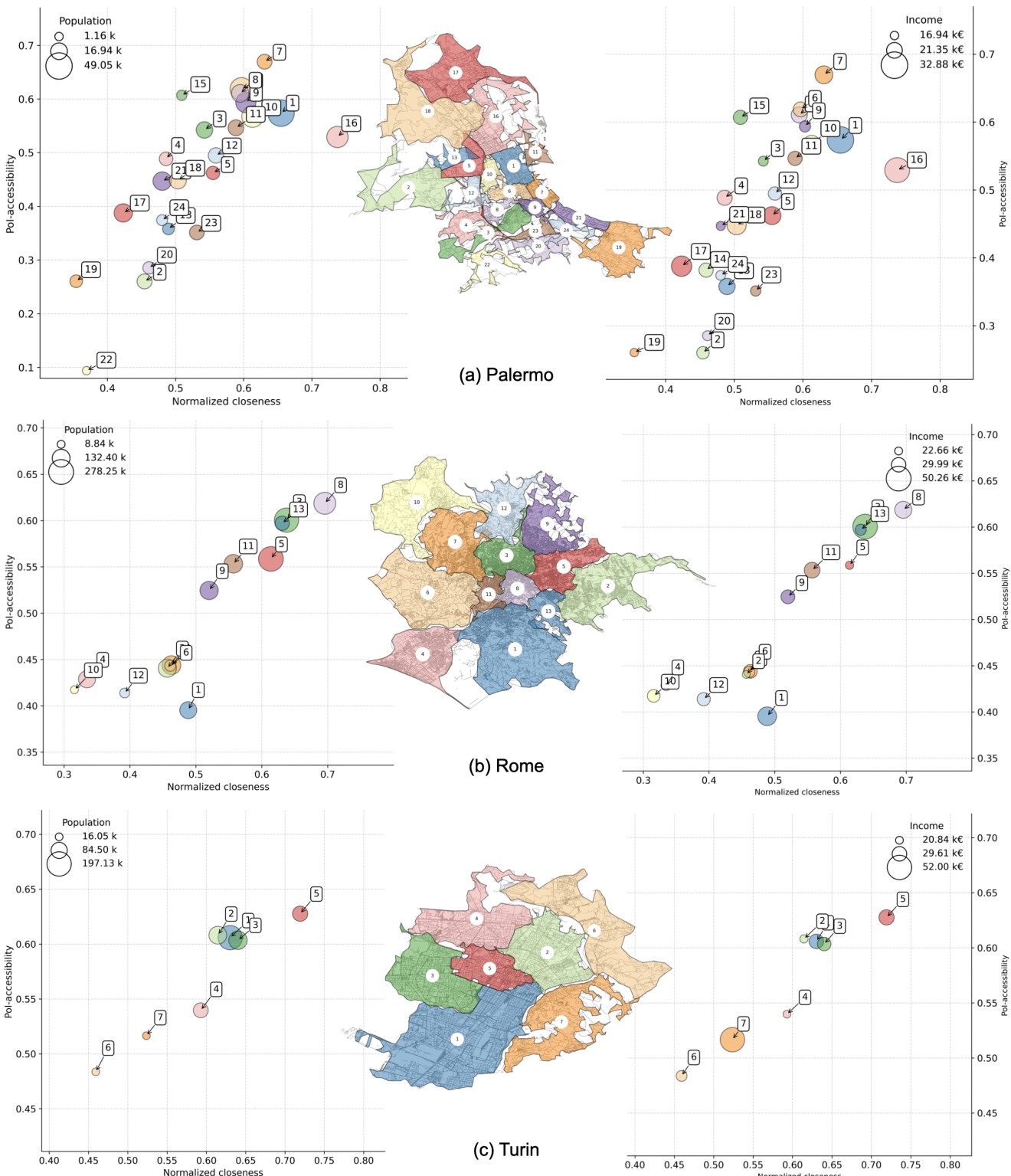

**Fig 16. Pol-accessibility vs closeness by neighborhoods in Palermo (a), Rome (b), and Turin (c).** Markers are color-coded according to the administrative neighborhoods (as shown in the embedded city map), and sized by population (left) and average income (right). Axes are ranged differently, to simplify the comparison between neighborhoods within a city. *Maps contain information from OpenStreetMap and OpenStreetMap Foundation, which is made available under the Open Database License.*

If we take a closer look at the most populated Italian cities, we can find a great heterogeneity that somehow mirrors what happens at a coarse-grained resolution. Southern Italian cities and Rome generally suffer from lower accessibility and poorer connectivity, while Northern Italian cities and Florence over-perform, even if compared with other top world-wide cities in the upper right region of Fig 10.

This picture becomes more nuanced when zooming in at the neighborhood level. Interestingly, we identify clusters in several cities where wealthier districts exhibit both low accessibility and weak normalized closeness — a potential signal of voluntary isolation, similar to gated communities described in other contexts. Such areas may rely heavily on private transportation, weakening the premise that wealthier urban environments naturally support the 15-minute city model. Milan stands out as an exception: here, PoI-accessibility and closeness are strongly correlated, and even values for lower-income clusters are close to the city median.

These results suggest that following the 15-minute city paradigm does not automatically translate into greater inclusiveness or reduced segregation.

## Discussion

Our analysis reveals that while proximity-based metrics provide a useful first picture of urban hyper-proximity, they capture only part of the complexity of service accessibility. By design, PoI-accessibility extends this view by integrating both the diversity (PoI-entropy) and density (PoI-density) of available services. This results in more nuanced insights, as reflected in both the city rankings and the internal spatial differences we observe. For instance, even cities ranking high by proximity, like Paris, still display significant within-city inequalities, with peripheral neighborhoods lacking access to essential services.

### Comparison with prior proximity-based studies

Comparing our results with prior studies using a proximity measure based on the average distance to services (as in [10,11,25]), we find that the city rankings produced by the different approaches are positively correlated with each other. Fig 17 shows the Kendall's Tau value of each ranking pair. Here, we observe that different measures of proximity to PoIs, calculated by independent scholars on data retrieved from OSM at different times, describe a comparable yet heterogeneous global pattern. Interestingly, our PoI-proximity metric leads to a ranking whose correlation with the measures used in [25] and in [11] (respectively 0.57 and 0.38) is stronger than the correlation between the two of them (i.e., 0.32), indicating that our approach captures aspects of both while also introducing distinctive differences. Similarly, this is the case for the city ranking generated by PoI-accessibility, a metric that integrates PoI-density and PoI-entropy dimensions to PoI-proximity, which also shows very strong correlations with ranks by $\mathcal{P}(n)$, and $\mathcal{P}_{\text{avg}}$. This addition of density and entropy, indeed, allows us to uncover important differences, especially in cases where services are clustered or overly specialized. More details on comparing rankings by PoI-accessibility, PoI-proximity and PoI-proximity-avg can be found in the Supplementary Material (S2 Appendix: More on cities' rankings).

When shifting focus to closeness, which measures network connectivity rather than local access, we see a radically different picture. The weak and statistically non-significant correlation between closeness and all other metrics highlights how this measure captures complementary — and largely independent — aspects of urban functioning. In other words, high accessibility to services does not necessarily imply good connectivity within the transport network. As a result, some well-served neighborhoods remain relatively disconnected, while others benefit from both strong service provision and better network connectivity.

Our analysis strengthens the argument that comparing PoI-proximity, PoI-entropy, PoI-density (combined in a single measure PoI-accessibility) and closeness, we have a better chance to capture the complexity of urban accessibility landscapes.

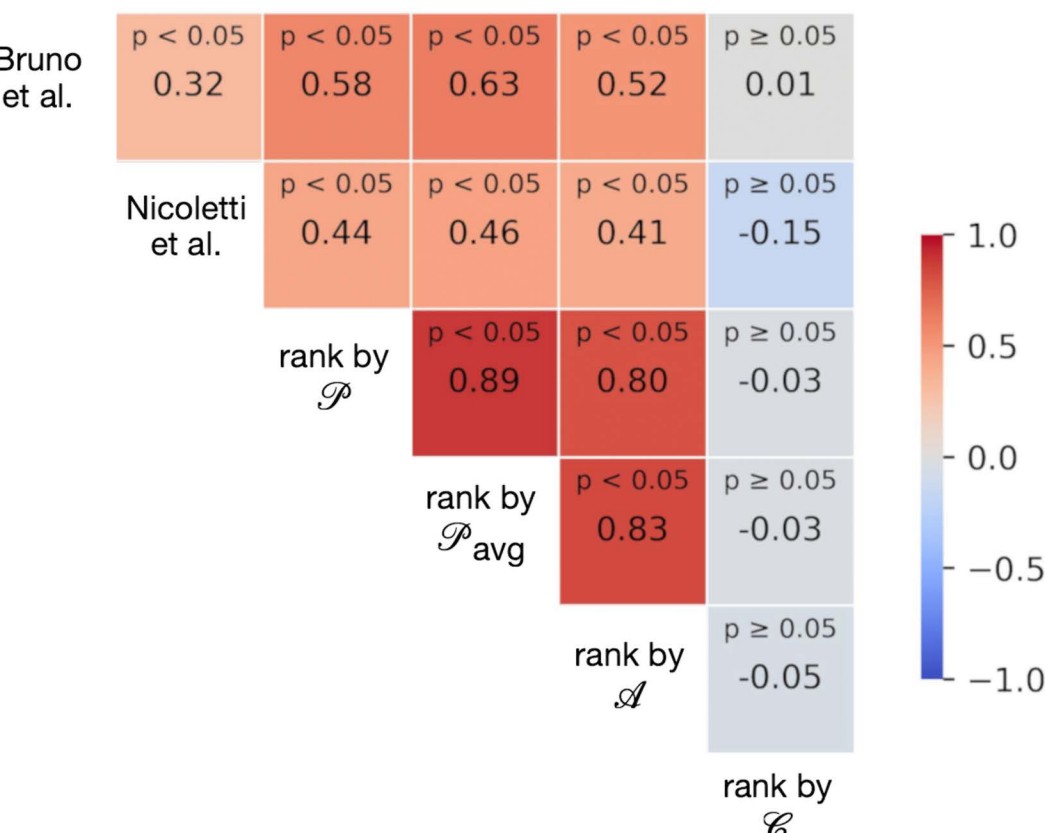

**Fig 17. Kendall's Tau correlation between different city rankings across studies.** Correlation between city rankings based on the accessibility measures presented in this paper, to Bruno et al. [25], and Nicoletti et al. [11]. The comparison is restricted to the subset of cities included in all three studies. Colors range from −1 (blue, strong negative correlation) to 1 (red, strong positive correlation), with grey indicating near-zero correlation. Reported values indicate the corresponding correlations and p-values.

**Final notes on the socio-demographic analysis.** This analysis challenges the 15-minute city paradigm's inclusivity assumption, revealing structural exclusion in low $P_n$ > 15 min and closeness-deficient lower-income neighborhoods (e.g., 40% Naples residents) [42]. These patterns align with spatial justice theories where transport inequities perpetuate stratification [18,33,43]. Urban science insights—from global rankings to domestic disparities—underscore agency in affluent isolation while urging equity-aware metrics over proximity-alone planning to prevent socio-economic masking.

**Policy applications.** Our multi-scale metrics offer actionable guidance for policymakers and urban planners to advance spatial justice and SDG 11. For zoning reforms, planners could prioritize mixed-use developments in low PoI-accessibility clusters (e.g., peripheral neighborhoods with Poi-Proximity higher than 15 min), mandating service quotas like one health PoI per km² to counter core-periphery imbalances, that has been observed in 70% of analyzed cities. In equitable service distribution, results suggest reallocating facilities—such as community spaces—from high-entropy centers to low-closeness postal codes in cities like Naples, where lower-income areas comprise up to 40% of poorly served residents, using our Infomap-detected clusters for targeted GTFS-integrated transit enhancements. For resilience planning, scenario simulations on the United-and-Close platform could model disruptions (e.g., bus line failures), recommending redundancy in bikable networks for voluntary isolation zones, such as the high-income Bologna districts, ensuring hyper-proximity withstands shocks while reducing segregation risks.

## Limitations and future perspectives

Despite the robustness of our approach, it is important to recognize certain inherent limitations. First, the study relies on OpenStreetMap as a primary data source for mapping pedestrian networks and Points of Interest. While OSM enables large-scale, open urban analysis, it also presents well-known limitations related to completeness and consistency across regions and PoI reporting. This data variability may influence local measurements of accessibility and service availability. Zhou et al. (2022) [44], for instance, discuss the limitations of volunteered geographic information for spatial analysis in urban contexts, showing that although many countries exhibit relatively low completeness, they often maintain a high degree of positional accuracy. Barrington-Leigh and Millard-Ball (2017) [45] uncovered the presence of regional disparities in data coverage, yet also found OSM to be about 80% complete globally, with over 40% of fully mapped countries at the street network level. This variability does not follow a simple Global North–Global South divide. Instead, OSM completeness differs widely within and across regions, income groups, and urban contexts. Factors such as governance, Internet access, data import policies, urban density, and targeted humanitarian or community mapping efforts play a significant role, while income alone has been shown to have a limited independent effect on completeness. As a result, differences observed across cities in our analysis may partly reflect localized data gaps or mapping saturation levels rather than systematic regional biases. Despite its limitations, the broad coverage and open nature of OSM has promoted its use in numerous accessibility and mobility studies (e.g., [11,25,46]. Future work should explore alternative or complementary datasets, such as Overture Maps (https://overturemaps.org/, last access: Jan. 22, 2026), to assess the robustness of findings and enhance data reliability.

Second, instead of using precise residential address data, we approximate population distribution using road intersections and WorldPop estimates. While this method does not provide household-level precision, it offers higher spatial granularity compared to grid-based approaches, which use the centroid of a cell as the origin of the isochrone and assign the entire cell's census population to that point (e.g., [11,25]). In contrast, intersections are inherently part of the street network, more accurately reflecting the surrounding environment and effectively estimating the local population distribution whether within census boundaries or, as in our study, within the WorldPop cell.

Third, there are public transport data constraints. When considering our urban connectivity analysis, we used GTFS data where available. However, public transit coverage and schedule accuracy differ across cities, potentially influencing closeness centrality calculations, especially for cities with missing or outdated GTFS data.

Finally, while our study leverages income data for sociodemographic validation in Italian cities, we acknowledge that this approach may limit the generalizability of our findings, especially in global contexts where such data are incomplete or unavailable. The reliance on income as a sole indicator of socioeconomic status can restrict the transferability of our methodology to cities in the Global South or other regions with less robust official statistics.

To address these challenges and enhance the generalizability and applicability of our approach, future research should consider integrating alternative proxies for socioeconomic status. Measures such as satellite-derived night-time light intensity, housing age, and land value [47,48] have proven to be effective in detecting patterns of urban segregation and economic activity, especially in contexts where income data are scarce or unavailable. Similarly, it is also important to note that informal and unregistered services, such as street vendors and unregulated transport networks, play a significant role in many lower-income areas but are not represented in our dataset. While integrating these elements is beyond the scope of the current study, future research should explore strategies to integrate these elements to provide a truly comprehensive assessment of service accessibility in such contexts.

Moving forward, we plan to integrate additional data sources, such as official municipal data, crowd-sourced updates, and alternative datasets on urban mobility and service distribution, to enhance data completeness. We will also explore complementary methods, including remote sensing and census data fusion, to improve the accuracy of residential distribution models. Incorporating dynamic mobility data, such as real-time transit schedules and pedestrian flow information, will allow us to better capture temporal variations in accessibility and connectivity.

A key next step is the integration of the multi-scale metrics outlined in this paper into an online platform, United-and-Close [38], designed to offer citizens and policymakers an interactive tool for data analysis and visualization across multiple cities globally. This platform will offer actionable insights for urban planners and policymakers seeking to promote more equitable and inclusive cities. By quantifying accessibility and connectivity at multiple scales, these metrics serve as diagnostic tools for identifying spatial disparities in service provision and transport integration. Planners can leverage this information to revise zoning practices, promote mixed-use developments, and ensure more balanced distribution of essential services across neighborhoods, thus counteracting the clustering of resources in already advantaged areas [49].

In the context of transit equity and resource allocation, recent work has proposed PoI reallocation algorithms to maximize 15-minute accessibility [25]. However, such large-scale redistribution is often impractical due to real-world constraints. This highlights the need for tools that identify where interventions are most needed. In this context, our framework and platform can: 1) enable the identification of under-served areas where public transport connectivity is lacking, guiding targeted investments in transit infrastructure and service improvements; 2) support the allocation of resources to neighborhoods where enhanced mobility can have the greatest impact on reducing social and economic exclusion [50]; and 3) inform participatory planning processes, empowering communities to advocate for interventions that address their specific needs and barriers to access.

Building on these functionalities, we will also evaluate urban accessibility resilience through scenario-based simulations. These will include disruption scenarios (e.g., service interruptions or network failures) to assess system robustness, and intervention scenarios (e.g., adding new services, cycling infrastructure, or transit stops) to estimate potential improvements in accessibility. This approach will provide policymakers with actionable, context-sensitive insights to support targeted enhancements without requiring large-scale restructuring of urban systems.

These opportunities position our platform as both an evaluative and strategic tool to support more inclusive and equitable urban development, in line with the 11th goal of the United Nations 2030 Agenda for Sustainable Development "Make cities and human settlements inclusive, safe, resilient and sustainable" (https://sdgs.un.org/goals/goal11, last access: Jan. 22, 2026) and consistent with the broader sociotechnical and decarbonization agendas that cities are increasingly expected to address as part of emerging urban models such as the 15-Minute City [51].

## Conclusions

This study leverages complex network analysis to quantify the interplay between urban segregation and service accessibility across 81 cities worldwide. Key findings reveal significant disparities: while 38 cities achieve average PoI-proximity under 15 minutes (Paris at 5.3 min, IQR 4.0–6.0, covering 99.7% of residents), lower-ranked cities like Houston show only 9.8% coverage with averages exceeding 36 minutes (IQR 26.0–47.0).

At neighborhood scale, PoI-accessibility and normalized closeness correlate strongly within cities (Pearson's $r$ from 0.62 in Ottawa to 0.98 in Melbourne), yet disadvantaged areas—often lower-income Italian postal codes—exhibit both poor accessibility (mean 0.35) and connectivity (0.42). High-income Bologna, Genoa, Naples, Palermo, Rome, and Turin clusters demonstrate voluntary isolation with low $P_n$ and closeness, signaling that hyper-proximity alone fails to ensure inclusivity, risking segregation exacerbation in up to 90% under-served residents.

## Supporting information

**S1 Appendix. Point of Interests' OSM categories S1_categories.pdf.**
(PDF)

**S2 Appendix. More on cities' rankings S2_rankings.pdf.**
(PDF)

**S3 Appendix. Distributions of $P(n)$, $D(15, n)$, $E(15,n)$, and $A(15,n)$ for all the other cities S3_distributions.pdf.**
(PDF)

**S4 Appendix. Cities' bubble charts by geographical region S4_bubblecharts.pdf.**
(PDF)

**S5 Appendix. Plots on the 5 representative cities at a greater resolution S5_5cities.pdf.**
(PDF)

**S6 Appendix. More details on Italian cities S6_10italiancities.pdf.**
(PDF)

**S7 Appendix. Comparing Closeness to *P*, *D*, and $\mathcal{E}$ S7_otherbubblecharts.pdf. Software: Open Source code and plots** https//github.com/mirkolai/cities To allow experiments replication, the source code to download data from sources, and calculate the accessibility and closeness measures is released under the GPL-3 License. At the same repo, we released all the plots used in the paper, in the Supplementary Material, and the others we could not include for the sake of brevity.
(PDF)

## Acknowledgments

The research that led to this paper started in 2022, when ML, SV and GR were at the University of Turin.

## Author contributions

**Conceptualization:** Mirko Lai, Federica Cena, Giancarlo Ruffo.

**Data curation:** Anna Sapienza.

**Formal analysis:** Anna Sapienza.

**Funding acquisition:** Giancarlo Ruffo.

**Investigation:** Mirko Lai.

**Methodology:** Mirko Lai, Salvatore Vilella, Federica Cena, Giancarlo Ruffo.

**Resources:** Anna Sapienza, Massimo Canonico.

**Software:** Mirko Lai, Salvatore Vilella, Massimo Canonico.

**Supervision:** Giancarlo Ruffo.

**Validation:** Anna Sapienza, Salvatore Vilella.

**Visualization:** Mirko Lai, Anna Sapienza, Giancarlo Ruffo.

**Writing – original draft:** Mirko Lai, Giancarlo Ruffo.

**Writing – review & editing:** Anna Sapienza, Federica Cena.

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
