## [Decision Letter · Decision Letter 0]

23 Jun 2025

Dear Dr. Ruffo,

Thank you for submitting your manuscript to PLOS ONE. After careful consideration, we feel that it has merit but does not fully meet PLOS ONE’s publication criteria as it currently stands. Therefore, we invite you to submit a revised version of the manuscript that addresses the points raised during the review process.

We look forward to receiving your revised manuscript.

Kind regards,

Abeer Elshater

Academic Editor

PLOS ONE

Journal Requirements:

[SV has been partially funded by PRIN 2022 (PNRR M4C2) EU Commission - Next Generation EU, Mission 4 Component 1 CUP C53D23005810006].

4. Thank you for stating the following in your manuscript:

[This research has been partially funded by PON “Ricerca e Innovazione” 2014-2020 716 framework, and by the European Union - Next Generation EU, Mission 4 Component 2 717 - CUP C53D23005810006, and CUP C33C24000340001.]

[SV has been partially funded by PRIN 2022 (PNRR M4C2) EU Commission - Next Generation EU, Mission 4 Component 1 CUP C53D23005810006]

[NO authors have competing interests].

6. We noted in your submission details that a portion of your manuscript may have been presented or published elsewhere.

[Fig. 2 has been used also in paper "A Complex Networks Approach to Evaluate the 15-Minute City Paradigm and Urban Segregation at Different Scales", to be published in the Proceedings of the International Conference on Complex Networks and Their Applications, 2024. The present submission has been invited to extend that work.]

Please clarify whether this conference proceeding was peer-reviewed and formally published. If this work was previously peer-reviewed and published, in the cover letter please provide the reason that this work does not constitute dual publication and should be included in the current manuscript.

7. We note that Figures 1, 6, 7, 8, 11, 13, 14, 15, 16, Appendices S5 and S6 in your submission contains a map image which may be copyrighted. All PLOS content is published under the Creative Commons Attribution License (CC BY 4.0), which means that the manuscript, images, and Supporting Information files will be freely available online, and any third party is permitted to access, download, copy, distribute, and use these materials in any way, even commercially, with proper attribution. For these reasons, we cannot publish previously copyrighted maps or satellite images created using proprietary data, such as Google software (Google Maps, Street View, and Earth). For more information, see our copyright guidelines: http://journals.plos.org/plosone/s/licenses-and-copyright.

1. You may seek permission from the original copyright holder of Figures 1, 6, 7, 8, 11, 13, 14, 15, 16, Appendices S5 and S6 to publish the content specifically under the CC BY 4.0 license.

Reviewers' comments:

Reviewer's Responses to Questions

**Comments to the Author**

1. Is the manuscript technically sound, and do the data support the conclusions?

Reviewer #1: Partly

Reviewer #2: Yes

2. Has the statistical analysis been performed appropriately and rigorously?

Reviewer #1: Yes

Reviewer #2: Yes

3. Have the authors made all data underlying the findings in their manuscript fully available?

Reviewer #1: Yes

Reviewer #2: Yes

4. Is the manuscript presented in an intelligible fashion and written in standard English?

Reviewer #1: Yes

Reviewer #2: Yes

Reviewer #1: Dear authors,

This study is rich and comprehensive in scope, as it examined 81 cities worldwide, providing broad insights rather than focusing on a single location. However, there are data limitations: while openstreetmap and other public datasets are valuable, they may lack uniform coverage across all cities, potentially affecting accuracy. It is essential to discuss how these limitations could potentially skew the results or lead to misinterpretations of urban density and accessibility. A comparative analysis or a discussion on the reliability of these databases in urban studies would enhance the credibility of your findings.

Also, the shift from the understanding of density in terms of population per square kilometer to focusing on the density of Points of Interest (POIS) within each isochrone is a significant methodological choice that requires further justification (line 177). To the assumption that road network intersections can serve as proxies for population locations (lines 88-94) is a substantial leap that demands a stronger theoretical foundation. It would be beneficial to situate this decision within existing literature, providing examples of previous studies that have successfully employed similar methodologies. This will address the concerns about accuracy and strengthen the argument for the methodological choices made as well.

Considering the diverse range of cities studied, an in-depth analysis of how socio-cultural dynamics shape urban accessibility is essential.

It is commendable that the methodology and visual results are made publicly available on github. However, the accessibility and functionality of the provided links must be verified.

It should be noted that listing findings within the introduction section is unconventional and potentially confusing for the reader. The introduction should set the context, define the research gap, and outline the objectives and significance of the study without delving into the results.

Moreover, the manuscript does not clearly articulate how categories such as Health and well-being are evaluated within the study. Providing a detailed methodology on the evaluation criteria and process will enhance the transparency and reproducibility of the research.

Reference needs to include recent studies; recheck the reference formatting as in (Bruno M, Monteiro Melo HP, Campanelli B, Loreto V. A universal framework for inclusive 15-minute cities. Nature Cities. 2024;1:633–641.) this reference is incomplete.

In line 103, please mention which supplementary material you are referring to.

Lastly, avoid using jargon such as "ghettoization" without providing a definition, and if "segregation" suffices, refrain from overusing it. Alternatively, on page 10, line 18, the phrase "HERE, WE introduce" needs more formal technical writing. On page 30, line 595, a clearer trend emerges. Avoid including extraneous comments such as "Taken together" in the abstract.

Reviewer #2: This manuscript offers a socially relevant and technically rigorous investigation that examines the relationship between socio-spatial segregation and urban service accessibility within the context of the 15-minute city paradigm. The authors implement a robust methodology that is based on network science and utilises PoI proximity, density, entropy, and closeness centrality in 92 global cities.

Furthermore, it emphasises the study's methodological coherence and innovation. The authors offer a comprehensive comprehension of the ways in which infrastructural accessibility does not always translate into equitable connectivity by integrating node-based accessibility metrics with normalised closeness centrality within a multi-scale spatial framework. This effectively expands upon previous computational urbanism research (e.g., Hillier & Hanson, 1984; Porta et al., 2006) while also building upon the scope of earlier endeavours such as Bruno et al. (2020) and Nicoletti et al. (2021). The decision to model city clusters using Infomap instead of administrative boundaries is particularly commendable, as it provides a more naturalistic representation of urban morphology.

Furthermore, the investigation is exceptional in its transparency and openness to scientific inquiry. The authors have granted complete access to their code repositories, which encompass scripts for data retrieval, processing, and visualisation. The methodological credibility is improved by the clear description of the use of cloud-based batch processing for scaling computational tasks across large networks.

However, there are a few areas that could benefit from specific improvements:

Limitations of Socioeconomic Coverage: Although the sociodemographic validation of Italian cities by income data is valuable, the conclusions regarding urban segregation may appear to be restricted in their generalisability. The authors are encouraged to recognise the potential of alternative proxies (e.g., night-time light data, housing age, or land value, as demonstrated in Bilal et al., 2019; Venerandi et al., 2018) in situations where income data are not readily accessible. This would enhance the paper's relevance to cities in the Global South that have incomplete datasets.

Clarification of Terminology: The manuscript could benefit from a more distinct differentiation between network-based disconnection and spatial segregation. Although closeness centrality is a valuable proxy for topological isolation, it may not fully capture the sociopolitical aspects of urban exclusion. Interpretive nuance could be improved by contextualising this within the context of established urban inequality literature (e.g., Lees, 2017; Lucas, 2012).

Comparative Literature Positioning: In addition to comparisons with Nicoletti and Bruno, this study could be more clearly positioned within the emerging global discourse on walkability and equity by including references to recent work on accessibility-diversity metrics (e.g., Kim & Kang, 2020; Welle et al., 2018).

Policy Implication Framing: Although the empirical results are robust, the discussion could be enriched to provide a more precise explanation of the implications for urban planning. In particular, how could planners or policymakers employ these metrics to revise zoning practices, promote transit equity, or prevent service clustering? The framework is consistent with the call for spatial justice and should be more explicitly linked to such agendas.

Visual Simplification: A few figures, particularly bubble charts and heatmaps, are dense and could be more clearly annotated. Highlighting extreme or illustrative cases (e.g., Paris, Houston, Jakarta) would assist in interpretation. A comparative schematic explaining how PoI proximity, density, and entropy relate conceptually to accessibility versus closeness would also aid clarity.

Language and Presentation: The manuscript is generally well-written. A few minor edits are required to ensure that Figure captions are clear. These issues may be resolved through revision.

**Do you want your identity to be public for this peer review?** For information about this choice, including consent withdrawal, please see our Privacy Policy

Reviewer #1: **Yes:** Mirame Elsayed

Reviewer #2: **Yes:** SAYON PRAMANIK

---

## [Author Response · Author response to Decision Letter 1]

6 Aug 2025

Editor' comments:

We checked the coherence of our manuscript with PLOS ONE’s style requirements and edited it as follows: 1) we updated the formatting of authors names and affiliations; 2) we included the corresponding author line; 3) we changed the file extension of figures and file naming according to PLOS ONE’s convention.

We corrected the information about Fundings and Financial disclosure.

[SV has been partially funded by PRIN 2022 (PNRR M4C2) EU Commission - Next Generation EU, Mission 4 Component 1 CUP C53D23005810006].

We followed your instructions in our new submission.

4. Thank you for stating the following in your manuscript:

[This research has been partially funded by PON “Ricerca e Innovazione” 2014-2020 716 framework, and by the European Union - Next Generation EU, Mission 4 Component 2 717 - CUP C53D23005810006, and CUP C33C24000340001.]

[SV has been partially funded by PRIN 2022 (PNRR M4C2) EU Commission - Next Generation EU, Mission 4 Component 1 CUP C53D23005810006]

According to the Editor’s points 2, 3, and 4 we have removed incorrect funding information from the manuscript and included the amended statement in the cover letter. Our final financial statement is the following:

This research has been partially funded by PON “Ricerca e Innovazione” 2014-2020 716 framework, and by the European Union - Next Generation EU, Mission 4 Component 2 717 - CUP C53D23005810006, and CUP C33C24000340001. There was no additional external funding received for this study.

[NO authors have competing interests].

6. We noted in your submission details that a portion of your manuscript may have been presented or published elsewhere. Please clarify whether this conference proceeding was peer-reviewed and formally published. If this work was previously peer-reviewed and published, in the cover letter please provide the reason that this work does not constitute dual publication and should be included in the current manuscript.

[Fig. 2 has been used also in paper "A Complex Networks Approach to Evaluate the 15-Minute City Paradigm and Urban Segregation at Different Scales", to be published in the Proceedings of the International Conference on Complex Networks and Their Applications, 2024. The present submission has been invited to extend that work.]

Please clarify whether this conference proceeding was peer-reviewed and formally published. If this work was previously peer-reviewed and published, in the cover letter please provide the reason that this work does not constitute dual publication and should be included in the current manuscript.

Thank you for raising this point.

We did not anticipate this to be an issue, as the current submission was prepared following an invitation to submit an extended version of our previous work, “A Complex Networks Approach to Evaluate the 15-Minute City Paradigm and Urban Segregation at Different Scales” , for publication in your journal.

Figure 2 in the original submission had also appeared in that earlier work, which meanwhile has been published in the Proceedings of the International Conference on Complex Networks and Their Applications, held in Istanbul in 2024. However, in light of the potential licensing concerns this might raise, we have chosen to remove the figure from the manuscript, even though it was initially helpful for context. We believe the article remains clear and self-contained without it. We also added a reference to the Stephen Few classic textbook in data visualization “Now You See It: Simple Visualization Techniques for Quantitative Analysis” that contains an extended description of bubble plots as extension of scatter plots.

7. We note that Figures 1, 6, 7, 8, 11, 13, 14, 15, 16, Appendices S5 and S6 in your submission contains a map image which may be copyrighted. All PLOS content is published under the Creative Commons Attribution License (CC BY 4.0), which means that the manuscript, images, and Supporting Information files will be freely available online, and any third party is permitted to access, download, copy, distribute, and use these materials in any way, even commercially, with proper attribution. For these reasons, we cannot publish previously copyrighted maps or satellite images created using proprietary data, such as Google software (Google Maps, Street View, and Earth). For more information, see our copyright guidelines: http://journals.plos.org/plosone/s/licenses-and-copyright.

1. You may seek permission from the original copyright holder of Figures 1, 6, 7, 8, 11, 13, 14, 15, 16, Appendices S5 and S6 to publish the content specifically under the CC BY 4.0 license.

We addressed these important concerns with the following actions.

1. The only figures that include OSM map tiles in the manuscript are Figures 1, and 7 (we use current figures numbering here). We thus modified the captions according to PLOS One guidelines (https://journals.plos.org/plosone/s/figures#loc-maps) stating that "OpenStreetMap map tiles are free to use as long as they are accompanied by the following attribution statement: “Base map and data from OpenStreetMap and OpenStreetMap Foundation”. Maps created using OpenStreetMap data must be accompanied by the following attribution statement: "Contains information from OpenStreetMap and OpenStreetMap Foundation, which is made available under the Open Database License.” We updated Fig. 1 and 7 captions accordingly.

2. All other figures displaying city maps (Figures 6, 8, 11, 12, 14, 15, and 16, we use current figures numbering), as well as the figures contained in S5, S6 and the novel S7, do not actually use any map tiles. Instead, we use the coordinate data to project intersection and street data points on a Cartesian plane thus not encountering the issue of using maps under the OSM CC BY-SA 2.0. We updated all of the related figures' captions accordingly.

3. We further queried the PLOS One editorial staff if the previous two actions suffice, and on July 17th we received the following answer from Zora Nazarei, Peer Review Operations Specialist:

“Before we proceed, we need some additional information to ensure the images contained in your submission comply with PLOS ONE copyright policy (https://journals.plos.org/plosone/s/licenses-and-copyright).

PLOS ONE requires direct confirmation that all images contained in submissions can be published under the CC BY 4.0 license utilized by PLOS ONE. This license allows unrestricted reproduction and reuse (even commercial) of images published in PLOS.

In response to your query above, Open StreetMaps is an acceptable source and does comply with our CC BY 4.0 license.

Please respond to the following query regarding the maps/satellite images contained in Figures 1, 6, 7, 8, 11, 13, 14, 15, 16, Appendices S5 and S6:

- Where did the authors obtain the maps, basemaps, shapefiles, map data, etc in these figures?

- What software was used to make these maps?

Please provide the direct URL for the relevant webpage or webpages where your map data was retrieved. Please ensure that licensing information for the relevant data is present on the webpage(s) you provide.

Please also update the captions for Figures 1, 6, 7, 8, 11, 13, 14, 15, 16, Appendices S5 and S6 to include any required attribution associated with the applicable license or data source.

If we cannot confirm that your map was made using only publicly available data compatible with the CC BY 4.0 license, your map will need to be removed or replaced. If you choose to replace your maps, please update the relevant Figure captions, and provide us with information about your replacement map's data, including license information through the Editorial Manager system.

When we receive your response through Editorial Manager, we will evaluate the license associated with your map and confirm it is compatible with the CC BY 4.0 license utilized by PLOS ONE. Please be assured that we will reach out again at that time if there are any remaining issues with your submission.“

Then, we acknowledge that - as stated in the answer reported above - Open StreetMaps is an acceptable source and does comply with our CC BY 4.0 license. Finally, we reply below to the additional questions posed by the Peer Review Operations Specialist:

- Where did the authors obtain the maps, basemaps, shapefiles, map data, etc in these figures?

Map data, from all the figures in the manuscript and in the Supporting Information files (S5, S6 and S7), is from Open Street Map, as clarified above. Only Fig. 1 and 7 also use OSM map tiles. For all the other figures, we use the coordinate data to project intersection and street data points on a Cartesian plane, using our own software relying on the libraries listed below. All the figures’ captions have been updated to make the correct references to OSM data.

- What software was used to make these maps?

The software that we used is available at https://github.com/mirkolai/cities and make use of the following python libraries to process and visualize geo spatial data:

osmnx https://osmnx.readthedocs.io/en/stable/

pandas https://pandas.pydata.org/

geopandas https://geopandas.org/en/stable/

matplotlib https://matplotlib.org/

folium https://python-visualization.github.io/folium/latest/ (Fig.1 and 7 only, to visualize OSM map tiles)

We updated the Data section accordingly. It is our understanding that our usage of OSM data is permitted.

Reviewers' comments:

Reviewer #1:

Dear authors,

This study is rich and comprehensive in scope, as it examined 81 cities worldwide, providing broad insights rather than focusing on a single location.

However, there are data limitations: while openstreetmap and other public datasets are valuable, they may lack uniform coverage across all cities, potentially affecting accuracy. It is essential to discuss how these limitations could potentially skew the results or lead to misinterpretations of urban density and accessibility. A comparative analysis or a discussion on the reliability of these databases in urban studies would enhance the credibility of your findings.

We agree that while OpenStreetMap (OSM) and other public datasets enable large-scale comparative urban analysis, they are not without limitations—particularly with respect to consistency and completeness across different regions.

To address this, we have expanded the Limitations section of the manuscript to explicitly acknowledge potential coverage and quality variations in OSM data, which may influence the measurement of accessibility and the representation of urban density in certain cities.

We believe this additional discussion improves the transparency of our methodology and clarifies the potential sources of variability in our findings.

Also, the shift from the understanding of density in terms of population per square kilometer to focusing on the density of Points of Interest (POIS) within each isochrone is a significant methodological choice that requires further justification (line 177).

Thank you for pointing this out.

To clarify this point, we have expanded the manuscript to provide a stronger justification for our methodological choice and included a paragraph that elaborates on our conceptual framing.

Additionally, we have replaced the terms ‘proximity’, ‘density’, ‘entropy’, and ‘accessibility’

---

## [Decision Letter · Decision Letter 1]

1 Dec 2025

Dear Dr. Ruffo,

Thank you for submitting your manuscript to PLOS ONE. After careful consideration, we feel that it has merit but does not fully meet PLOS ONE’s publication criteria as it currently stands. Therefore, we invite you to submit a revised version of the manuscript that addresses the points raised during the review process.

We look forward to receiving your revised manuscript.

Kind regards,

Genyu Xu, Ph.D.

Academic Editor

PLOS ONE

Journal Requirements:

Reviewer's Responses to Questions

**Comments to the Author**

Reviewer #2: All comments have been addressed

Reviewer #3: (No Response)

Reviewer #4: (No Response)

2. Is the manuscript technically sound, and do the data support the conclusions?

Reviewer #2: Yes

Reviewer #3: Yes

Reviewer #4: Yes

3. Has the statistical analysis been performed appropriately and rigorously?

Reviewer #2: Yes

Reviewer #3: Yes

Reviewer #4: Yes

4. Have the authors made all data underlying the findings in their manuscript fully available?

Reviewer #2: Yes

Reviewer #3: Yes

Reviewer #4: Yes

5. Is the manuscript presented in an intelligible fashion and written in standard English?

Reviewer #2: Yes

Reviewer #3: Yes

Reviewer #4: Yes

Reviewer #2: This manuscript presents a strong and timely contribution to urban studies by investigating the interplay between accessibility to services and segregation within the framework of the “15-minute city.” The multi-scalar, network-based approach, which combines measures of Points of Interest (PoI) proximity, density, entropy, and closeness centrality across 92 cities worldwide, is a major strength. The methodological innovation, particularly the use of community detection and transport connectivity analysis, advances computational urbanism and contributes to the literature on spatial justice. The authors’ efforts to improve the manuscript based on earlier reviews are evident, and the expanded methodological clarity, updated figures, and broader literature engagement substantially strengthen the paper.

That said, a few areas merit further attention to enhance clarity, rigor, and impact:

Data Transparency and Limitations: While the manuscript now acknowledges the limitations of OpenStreetMap and other open datasets, further discussion on how heterogeneous data quality across regions may skew results would increase transparency. For instance, how do results differ for Global South cities where transport data is sparse compared to Europe or North America? A sensitivity check or comparison with alternative proxies (night-time light data, land value, housing age) could better demonstrate robustness.

Methodological Justification: The decision to treat intersections as population proxies has been clarified, but the argument would be stronger with more explicit references to prior studies that have validated this approach. Similarly, the weighting scheme for the combined accessibility score (equal weights to proximity, density, entropy) may benefit from a short justification or robustness test showing whether alternative weightings significantly affect outcomes.

Interpretive Nuance on Segregation: While closeness centrality is a useful proxy for topological isolation, it risks oversimplifying socio-spatial exclusion. The revised text acknowledges this, but the interpretation of “segregation” should be carefully phrased to avoid overextension. Integrating insights from the literature on urban inequality (e.g., Lees, Lucas, Soja) can help balance the structural-network perspective with the broader social dimension.

Results and Communication: The restructuring of figures and annotations has improved readability, but some visualizations (bubble charts, heatmaps) remain dense. Highlighting key illustrative cases such as Paris, Houston, or Jakarta, within the main text would make findings more accessible. The inclusion of a schematic overview of how the different accessibility metrics interrelate is very helpful and could be emphasized more clearly in the discussion.

Policy Relevance: The discussion has been expanded to touch upon spatial justice and SDG 11, which is commendable. Still, the paper would benefit from more concrete guidance for policymakers and urban planners. For example, how might results inform zoning reforms, equitable distribution of public services, or resilience planning? Providing specific examples or scenarios would strengthen the applied contribution of the work.

Language and Style: The manuscript is generally well-written, but a final round of language polishing would improve flow and readability. Avoid jargon such as “ghettoization” (as already addressed) and maintain a consistent formal tone. Some sentences in the abstract and introduction can be streamlined to avoid repetition.

Reviewer #3: This study investigates the relationship between walkable accessibility to services and urban transport connectivity across 92 global cities, filling a critical gap at both practical and theoretical levels: while the 15-minute city has been widely promoted as a sustainable urban planning strategy, its potential impacts on spatial segregation have remained insufficiently explored. The research methodology is generally rigorous and innovative, avoiding the limitations of single-indicator assessments. After a round of revisions, the logic and methodology of the article have become more refined, and it is no longer possible for me to identify major revisions that have not yet been proposed within the limited review time.

Although this article may still face some structural and fundamental questions, such as the validity of using 15 minute cities to measure segregation and equality, and the reliability of OSM data as an international comparison, whether the value judgment of segregation is universally applicable, and so on. However, the existence of these limitations is acceptable in an independent paper.

All maps in one figure should provide a scale reference, or in the same scale. Because 1. the urban scale may have an impact on the conclusions; 2. In Figures 6, 7, and 8, all the displayed cities appear to have similar scales. At this point, the "display weights" of analysis units with the same granularity in different cities may vary significantly, leading to a greater risk of misreading.

Although this study has innovated in both urban science and computer technology, demonstrating good process and technology, the analysis of its results is somewhat insufficient. I hope to include a separate chapter in the conclusion section, or before, to present the sociological or urban science conclusions of this study after completing international and domestic comparisons. The existing conclusions are scattered in the analysis of various chapters and tables, resulting in poor readability.

Reviewer #4: Manuscript needs revision in structure as follows

1. Kindly add separate section on the objectives of the study after introduction section clearly stating the objectives of the study along with scope and limitations.

2. In section on methods add information clearly stating different stages of research and giving information on methods of data collection and analysis to achieve the objectives of study, support it with figure also.

3. Support conclusion with data from Analysis section also do not add citations in conclusion section

**Do you want your identity to be public for this peer review?** For information about this choice, including consent withdrawal, please see our Privacy Policy

Reviewer #2: **Yes:** SAYON PRAMANIK

Reviewer #3: **Yes:** YE Nanqi

Reviewer #4: **Yes:** Tejwant Singh Brar

---

## [Author Response · Author response to Decision Letter 2]

16 Jan 2026

We attached file Response to Reviewers_v2.pdf to this submission

---

## [Editor Report · Decision Letter 2]

19 Jan 2026

Understanding the interplay between urban segregation and accessibility to services with network analysis

PONE-D-25-16347R2

Dear Dr. Ruffo,

We’re pleased to inform you that your manuscript has been judged scientifically suitable for publication and will be formally accepted for publication once it meets all outstanding technical requirements.

Kind regards,

Genyu Xu, Ph.D.

Academic Editor

PLOS One
---

## [Editor Report · Acceptance letter]

PONE-D-25-16347R2

PLOS One

Dear Dr. Ruffo,

I'm pleased to inform you that your manuscript has been deemed suitable for publication in PLOS One. Congratulations! Your manuscript is now being handed over to our production team.

Kind regards,

on behalf of

Dr. Genyu Xu

Academic Editor

PLOS One